# Interpretable Discovery of One-parameter Subgroups: A Modular Framework for Elliptical, Hyperbolic, and Parabolic Symmetries

**Pavan Karjol** [1]  **Vivek V Kashyap** [1]  **Rohan Kashyap** [2]  **Prathosh A P** [1]

## Abstract

We propose a modular, data-driven framework for jointly learning unknown functional mappings and discovering the underlying one-parameter symmetry subgroup governing the data. Unlike conventional geometric deep learning methods that assume known symmetries, our approach identifies the relevant continuous subgroup directly from data. Our framework focuses on three primary geometric components of one-parameter subgroup actions: elliptic, hyperbolic, and parabolic regimes. For the given regime, our framework instantiates a corresponding symmetry discovery architecture with invariant and equivariant representation layers structured according to the Lie algebra of the subgroup, and learns the exact generator parameters end-to-end from data. This yields models whose invariance or equivariance is guaranteed by construction and admits formal proofs, enabling symmetry to be explicitly traced to identifiable components of the architecture. The approach is applicable to one-parameter subgroups of a wide range of matrix Lie groups, including $SO(n)$, $SL(n)$, and the Lorentz group. Experiments on synthetic and real-world systems, including moment of inertia prediction, double-pendulum dynamics, and high-energy *Top Quark Tagging*, demonstrate accurate subgroup recovery and strong predictive performance across both compact and non-compact regimes.

## 1. Introduction

Invariance and equivariance provide powerful inductive biases in neural network design, leading to improved gener-alization, data efficiency, and robustness. This principle underlies a broad class of geometric deep learning models, including group-equivariant convolutional networks (Cohen & Welling, 2016), harmonic networks (Worrall et al., 2017), and $SE(3)$-transformers (Fuchs et al., 2020), which hard-code known symmetries into their architectures and have achieved strong performance across physical and geometric learning tasks.

In many scientific and engineering applications, however, the relevant symmetries are unknown or only partially specified. Imposing an incorrect or overly restrictive symmetry group can hinder learning, motivating the development of data-driven methods that infer underlying continuous symmetries directly from observations rather than assuming them a priori.

A particularly fundamental class of continuous symmetries is given by one-parameter subgroups: connected one-dimensional Lie subgroups generated by the exponential of a single Lie algebra element. Such subgroups capture elementary continuous transformations. In this work, we focus on three primary geometric components that arise in real block decompositions of linear actions: elliptic components, corresponding to compact rotational motion; hyperbolic components, corresponding to non-compact squeeze or boost-type motion; and parabolic components, corresponding to shear-type motion.

These regimes arise naturally in many matrix Lie groups, including $SO(n)$, $SL(n)$, and the Lorentz group (Hall, 2015; Helgason, 1978; Knapp, 2002).

In this work, we propose a modular, data-driven framework for the joint task of learning unknown functional mappings and discovering the underlying one-parameter symmetry subgroup governing the data. We emphasize that the geometric regime, elliptical, hyperbolic, or parabolic, is treated as a modeling assumption rather than a learned variable. Once a regime is assumed, the corresponding symmetry discovery framework, defined by a regime-specific invariant construction, is deployed, and the learning task focuses on recovering the exact subgroup parameters, including the generator and its associated structure, directly from data.

The core contribution of our work lies in the development

[1]Department of Electrical Communication Engineering, Indian Institute of Science, Bengaluru, Karnataka, India [2]Computer Science, Carnegie Mellon University, Pittsburgh, Pennsylvania, USA. Correspondence to: Pavan Karjol <pavankarjol@iisc.ac.in>.

*Proceedings of the 43$^{rd}$ International Conference on Machine Learning*, Seoul, South Korea. PMLR 306, 2026. Copyright 2026 by the author(s).

of *learnable, parameterized invariant representations* for one-parameter subgroups in the elliptic, hyperbolic, and parabolic regimes. For a fixed geometric regime, each invariant representation is explicitly constructed to be invariant under a specific 1-D subgroup action, while remaining *complete* in the sense that it separates orbits induced by that subgroup. The invariance properties are governed by a small set of learnable parameters, which directly encode the unknown Lie algebra generator through canonical block parameters and orientation.

As a direct consequence of these invariant constructions, we obtain symmetry discovery frameworks that share a common high-level architectural template. While the resulting models exhibit a unified form across regimes, this uniformity follows from the shared invariant–equivariant decomposition implied by our representations, rather than being the primary contribution. This structure enables the simultaneous, end-to-end recovery of both the subgroup parameters and the target functional mapping directly from data.

Crucially, the resulting models are **interpretable in a structural and analytical sense**. By interpretability, we mean that the source of invariance or equivariance is explicit in the architecture: these properties are enforced by the invariant representation layer and the learned subgroup parameters, rather than emerging only empirically. As a result, the symmetry properties admit formal mathematical proofs and can be directly attributed to identifiable components of the model. This stands in contrast to many existing symmetry discovery approaches (Ko et al., 2024; Benton et al., 2020; Moskalev et al., 2022; Shaw et al., 2024), where invariance emerges implicitly or is verified empirically, and therefore cannot be localized to specific architectural elements.

Although our main construction targets one-parameter subgroups, it also provides a modular primitive for higher-dimensional symmetry discovery: multiple one-parameter generators can be recovered through independent initializations or sequential orthogonalization, and then closed under Lie brackets to estimate a larger Lie algebra.

We validate the proposed framework across a diverse set of synthetic and real-world systems, spanning both compact and non-compact symmetry regimes, including moment of inertia prediction, double-pendulum dynamics, anisotropic quantum models, and high-energy *Top Quark Tagging*. Across these tasks, the framework consistently recovers meaningful one-parameter subgroup structures while maintaining strong predictive performance in both compact and non-compact regimes.

**Contributions**

- **Joint Learning of Functions and Symmetries:** We introduce a modular framework that simultaneously learns target functions and discovers their underlying one-parameter symmetry subgroups, enforcing invariance or equivariance as a structural inductive bias during training rather than via post-hoc discovery or data augmentation.

- **Unified Geometric Framework with Guarantees:** We provide a unified treatment of one-parameter subgroups across elliptical, hyperbolic, and parabolic regimes, and develop regime-specific invariant representation layers. We formally prove that these representations are both invariant and orbit-separating, ensuring expressive and complete symmetry encoding in both compact and non-compact settings.

- **Provable Structural Interpretability:** Our framework encodes symmetry directly into the architecture, yielding models whose invariance or equivariance is guaranteed by construction and admits formal proofs. As a result, symmetry can be explicitly attributed to identifiable components of the model and learned subgroup parameters, rather than emerging implicitly.

- **Empirical Validation Across Domains:** We validate the framework on a diverse set of synthetic and real-world systems, including moment of inertia prediction, double-pendulum dynamics, anisotropic quantum models, and high-energy *Top Quark Tagging*, demonstrating accurate subgroup recovery and strong predictive performance across all regimes.

## 2. Related work

### 2.1. Equivariance and Continuous Symmetries

Symmetry principles have become central to geometric deep learning (Bronstein et al., 2021). A foundational contribution is the formulation of $G$-equivariant neural networks (Cohen & Welling, 2016), which generalize convolutional architectures to known group actions. This framework was extended to arbitrary compact groups (Kondor & Trivedi, 2018) and homogeneous spaces (Cohen et al., 2019). Complementary to explicit equivariance, data augmentation has long been used to induce approximate invariance, particularly in vision (Simard et al., 2003; Krizhevsky et al., 2012; Perez & Wang, 2017).

A wide range of architectures have since been developed to realize equivariance across data modalities and symmetry groups. These include Euclidean-equivariant graph neural networks (Satorras et al., 2021), steerable CNNs based on equivariant filter bases (Weiler et al., 2018; Weiler & Cesa, 2019), Clifford algebra networks for orthogonal symmetries (Ruhe et al., 2023), and G-RepsNet for arbitrary matrix groups (Basu et al., 2024). A common limitation of these approaches is the assumption that the underlying symmetry group $G$ is known *a priori* and fixed during training.

*Table 1.* Comparison of symmetry discovery frameworks. We compare our method against Augerino (Benton et al., 2020), LieGAN (Yang et al., 2023), LieGG (Moskalev et al., 2022), InfGen (Ko et al., 2024), BeyondAffine (Shaw et al., 2024), and Bispectral Neural Networks (BNNs) (Sanborn et al., 2022) along four axes: whether symmetry handling is integrated into a *joint framework* during training, whether the prediction model has structurally interpretable invariance/equivariance, the scope of supported symmetries, and whether the method is used as an experimental baseline. Here, structurally interpretable means that the symmetry property is enforced by explicit architectural components and can be analytically verified.

| Method | Joint Framework | Interpretable Predictor | Scope | Baseline |
|---|---|---|---|---|
| **Ours** : $H_\gamma$-Net | **Yes** | **Yes** | 1D Lie groups | – |
| Augerino | **Yes** | No | Generic | ✓ |
| LieGAN | No (Discovery only) | No | Generic | – |
| LieGG | No (post-hoc) | No | Generic | – |
| InfGen | No (post-hoc) | No | Generic | – |
| BeyondAffine | No (post-hoc) | No | 1D Lie groups and non affine | – |
| BNNs | **Yes** | **Yes** | Discrete | – |

Continuous symmetries are naturally modeled by Lie groups, motivating extensions of equivariant learning to this setting. LieConv (Finzi et al., 2020) constructs convolutional layers equivariant to general Lie groups with surjective exponential maps, often relying on group discretization or truncated representation expansions (Weiler et al., 2018; Worrall et al., 2017). Clebsch–Gordan–based constructions were developed for $SO(3)$ (Kondor et al., 2018) and later generalized to arbitrary matrix groups (Finzi et al., 2021).

### 2.2. Automatic Symmetry Discovery

Recent work has focused on discovering symmetries when the transformation group is unknown. LieGG (Moskalev et al., 2022) extracts Lie group structure from trained networks post-hoc, while LieGAN (Yang et al., 2023) learns Lie algebra generators adversarially but decouples discovery from downstream prediction. Augerino (Benton et al., 2020) approximates invariance by learning distributions over transformations, BeyondAffine (Shaw et al., 2024) extends discovery beyond affine classes, and Bispectral Neural Networks (Sanborn et al., 2022) impose strong inductive biases via finite group approximations.

Most existing symmetry discovery methods primarily target compact groups and either operate post-hoc or do not integrate symmetry discovery with task-specific prediction in a unified, end-to-end framework. In contrast, our approach embeds symmetry discovery directly into the predictive architecture, enforcing invariance or equivariance by construction and enabling joint recovery of interpretable one-parameter subgroup generators. A comparative summary is provided in Table 1.

## 3. Proposed Method

Our framework leverages group-theoretic structure within a modular neural network architecture to jointly learn functional mappings and discover underlying continuous symmetries.

**Notation and background.** For completeness, a consolidated summary of notation (Section A) and relevant Lie-theoretic background (Section B) is provided in the Appendix. Readers may refer to this material as needed.

We consider the problem of learning an $H_\gamma$-invariant function $f : X \to \mathbb{R}^m$, where $X \subset \mathbb{R}^n$ is a $G$-set and $H_\gamma$ is an unknown one-parameter subgroup of a matrix Lie group $G$ acting on $X$. The function $f$ satisfies the invariance condition

$$f(h \cdot x) = f(x), \quad \forall h \in H_\gamma, \, \forall x \in X,$$

where $h \cdot x$ denotes the group action.

Our primary goal is to automatically discover the underlying symmetry subgroup $H_\gamma$ from data while jointly learning the associated invariant function $f$. This joint learning setting allows our framework to operate in scenarios where the governing symmetry is not known *a priori*, as commonly encountered in physical simulations, molecular modeling, and dynamical systems.

### 3.1. One-Parameter Subgroups and Orbit Classification

To formalize the structure of the symmetry subgroups $H_\gamma$ targeted by our learning framework, we briefly review the geometry of one-parameter Lie subgroups and the orbits they induce.

The Lie algebra $\mathfrak{g} = T_I G$ is the tangent space at the identity of a Lie group $G$. Each generator $B \in \mathfrak{g}$ defines a one-parameter subgroup via the homomorphism $\gamma : (\mathbb{R}, +) \to (G, \cdot)$ such that $\gamma(0) = I$ and $\gamma'(0) = B$. The exponential map $\exp : \mathfrak{g} \to G$, defined by $\exp(B) = \sum_{k=0}^{\infty} \frac{B^k}{k!}$, satisfies $\gamma(t) = \exp(tB)$, yielding the one-dimensional Lie subgroup (Hall, 2015; Varadarajan, 1984):

$$H_\gamma = \{\exp(tB) \mid t \in \mathbb{R}\} = \mathrm{Img}(\gamma). \tag{1}$$

Figure 1 provides a geometric visualization of a one-parameter subgroup as a smooth curve on the Lie group

generated by exponentiating a single Lie algebra element. We consider three fundamental geometric regimes arising from the canonical block decomposition $B = A^{-1}\hat{B}A$, where $A \in G$ determines the orientation and $\hat{B}$ is a canonical (e.g., real Jordan) form; these induce elliptic, hyperbolic, and parabolic one-parameter subgroups according to their orbit geometry (Knapp, 2002; Gantmacher, 1959; Helgason, 1978; Warner, 1983).

**Elliptical (Rotational) Regimes.** In the elliptical regime, the generator is conjugate to a block-diagonal matrix whose canonical form consists of skew-symmetric $2 \times 2$ blocks. The canonical generator takes the form

$$\hat{B}_{\text{ell}} = \bigoplus_{k=1}^{\lfloor n/2 \rfloor} \begin{pmatrix} 0 & \lambda_k \\ -\lambda_k & 0 \end{pmatrix} \oplus \mathbf{0}_{n \bmod 2},$$

$$\exp(t\hat{B}_{\text{ell}}) = \bigoplus_{k=1}^{\lfloor n/2 \rfloor} \begin{pmatrix} \cos(t\lambda_k) & \sin(t\lambda_k) \\ -\sin(t\lambda_k) & \cos(t\lambda_k) \end{pmatrix} \oplus I_{n \bmod 2},$$

$$(2)$$

where $\lambda_k \in \mathbb{Z}$ are discrete rotation rates, reflecting the restriction adopted in our symmetry discovery framework. The resulting one-parameter subgroup induces *closed orbits*.

**Hyperbolic (Squeeze / Boost) Regimes.** In the hyperbolic regime, the generator is real semisimple with nonzero eigenvalues occurring in $\pm$ pairs, giving rise to paired expanding and contracting directions. A canonical real $2 \times 2$ hyperbolic generator is

$$B_{\text{hyp}}(\lambda) = \begin{bmatrix} 0 & \lambda \\ \lambda & 0 \end{bmatrix}, \qquad \lambda \in \mathbb{R}, \tag{3}$$

whose exponential defines a hyperbolic transformation

$$\exp\big(tB_{\text{hyp}}(\lambda)\big) = H(t\lambda) := \begin{bmatrix} \cosh(t\lambda) & \sinh(t\lambda) \\ \sinh(t\lambda) & \cosh(t\lambda) \end{bmatrix} \tag{4}$$

In higher dimensions, the canonical generator and its exponential take the block-diagonal form

$$\hat{B}_{\text{hyp}} = \bigoplus_{k=1}^{\lfloor n/2 \rfloor} \begin{bmatrix} 0 & \lambda_k \\ \lambda_k & 0 \end{bmatrix} \oplus \mathbf{0}_{n \bmod 2},$$

$$\exp(t\hat{B}_{\text{hyp}}) = \bigoplus_{k=1}^{\lfloor n/2 \rfloor} H(t\lambda_k) \oplus I_{n \bmod 2}. \tag{5}$$

**Parabolic (Shear) Regimes.** A third class arises from nilpotent generators. In our framework, we restrict attention to 2-nilpotent generators, satisfying $B^2 = 0$, which are necessarily trace-zero and non-diagonalizable. The canonical

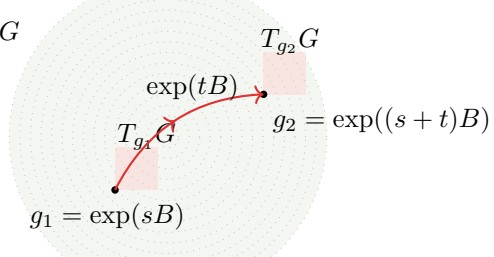

*Figure 1.* Visualization of one-parameter subgroups on Lie group $G$ through the exponential map.

generator takes the block-diagonal form

$$\hat{B}_{\text{par}} = \bigoplus_{k=1}^{\lfloor n/2 \rfloor} \begin{pmatrix} 0 & \lambda_k \\ 0 & 0 \end{pmatrix} \oplus \mathbf{0}_{n \bmod 2},$$

$$\exp(t\hat{B}_{\text{par}}) = \bigoplus_{k=1}^{\lfloor n/2 \rfloor} \begin{pmatrix} 1 & t\lambda_k \\ 0 & 1 \end{pmatrix} \oplus I_{n \bmod 2}. \tag{6}$$

The associated one-parameter subgroup induces shear-type orbits, which are non-periodic and exhibit polynomial drift.

*Consequently, discovering a one-parameter subgroup reduces to a modular optimization problem: learning the orientation $A$ and the canonical generator parameters $\{\lambda_k\}$ that specify $\hat{B}$ under the chosen regime.*

### 3.2. Generalized Invariant Representations

Building on the classification of one-parameter subgroups into elliptic, hyperbolic, and parabolic regimes, we now introduce a unified architectural mechanism for enforcing symmetry constraints within a neural network. The central goal is to construct a representation that is invariant along subgroup orbits while remaining maximally informative for downstream learning.

To this end, we introduce a modular *invariant representation* layer that maps each input $x$ to a canonical representative of its orbit under a one-parameter subgroup. This construction enforces invariance by collapsing each orbit to a unique representative while preserving all information relevant to the target function. These properties are formalized in Propositions 3.5 and 3.6, with proofs provided in Appendix C.

Throughout this section, we focus on the even-dimensional case $n$; the odd-dimensional extension requires only minor modifications and is deferred to Section H.3.

Let $x \in \mathcal{X} \subset \mathbb{R}^n$ and let $A \in GL(n)$ be a learnable orientation matrix. We express the input in subgroup-aligned

coordinates and decompose it into two-dimensional blocks.

$$v = Ax = \bigoplus_{i=1}^{n/2} v_i, \qquad v_i \in \mathbb{R}^2. \qquad (7)$$

The invariant representation is constructed by selecting a canonical element along the orbit of $x$. This is achieved by using the first block $v_1$ to solve for a subgroup parameter $t_0$, which aligns $v_1$ to a fixed reference configuration. The same transformation is then applied consistently across all blocks.

**Admissible domains:** The construction of the invariant representation requires excluding degenerate inputs for which the canonical alignment parameter $t_0$ is ill-defined. Accordingly, we introduce regime-specific admissible domains corresponding to the elliptic, hyperbolic, and parabolic cases.

Specifically, we define

$$\mathcal{X}_e := \{x \in \mathbb{R}^n \mid \|v_1\|_2 \neq 0\}, \qquad (8)$$
$$\mathcal{X}_h := \{x \in \mathbb{R}^n \mid N(v_1) \neq 0\}, \qquad (9)$$
$$\mathcal{X}_p := \{x \in \mathbb{R}^n \mid v_{1,2} \neq 0\}, \qquad (10)$$

where $v$ is as defined in equation 7 and

$$N(v_i) := \sqrt{|v_{i,1}^2 - v_{i,2}^2|}.$$

Each admissible domain excludes only a measure-zero subset of $\mathbb{R}^n$ and guarantees the existence and uniqueness of the alignment parameter $t_0$ in the corresponding regime.

**Notation.** Throughout Definitions 3.1–3.3, we use $v$ and its the block decomposition as introduced in equation 7.

**Definition 3.1** (Invariant Elliptic Representation: $\mathrm{invRep}_e$). Let $x \in \mathcal{X}_e$ and let $A \in GL(n)$ and $\lambda \in \mathbb{Z}^{n/2}$ parameterize an elliptic one-parameter subgroup. The invariant elliptic representation is defined as

$$\mathrm{invRep}_e(x) := \|v_1\|_2 \, \mathbf{e}_1 \; \oplus \; \bigoplus_{i=2}^{n/2} R(-t_0\lambda_i) \, v_i, \qquad (11)$$

where $t_0 \in \mathbb{R}$ is the unique parameter satisfying

$$R(t_0\lambda_1) \, \mathbf{e}_1 = \frac{v_1}{\|v_1\|_2},$$

and $R(\theta)$ denotes the 2D rotation matrix.

**Definition 3.2** (Invariant Hyperbolic Representation: $\mathrm{invRep}_h$). Let $x \in \mathcal{X}_h$, and let $A \in GL(n)$ and $\lambda \in \mathbb{R}^{n/2}$ parameterize a hyperbolic one-parameter subgroup. The invariant hyperbolic representation is defined as:

$$\mathrm{invRep}_h(x) := N(v_1) \, \mathbf{e} \; \oplus \; \bigoplus_{i=2}^{n/2} H(-t_0\lambda_i) \, v_i, \qquad (12)$$

where the canonical basis $\mathbf{e}$ and gauge parameter $t_0$ are defined according to the sector of $v_1$:

$$\begin{cases} \mathbf{e} = \mathrm{sgn}(v_{1,1}) \, e_1, \\ \tanh(t_0\lambda_1) = v_{1,2}/v_{1,1} \end{cases} \quad \text{if } |v_{1,1}| > |v_{1,2}|, $$
$$\begin{cases} \mathbf{e} = \mathrm{sgn}(v_{1,2}) \, e_2, \\ \tanh(t_0\lambda_1) = v_{1,1}/v_{1,2} \end{cases} \quad \text{if } |v_{1,2}| > |v_{1,1}|. \qquad (13)$$

**Definition 3.3** (Invariant Parabolic Representation: $\mathrm{invRep}_p$). Let $x \in \mathcal{X}_p$ and let $A \in GL(n)$ and $\lambda \in \mathbb{R}^{n/2}$ parameterize a parabolic one-parameter subgroup. The invariant parabolic representation is defined as

$$\mathrm{invRep}_p(x) := v_{1,2} \, \mathbf{e}_2 \; \oplus \; \bigoplus_{i=2}^{n/2} G(-t_0\lambda_i) \, v_i, \qquad (14)$$

where $t_0 \in \mathbb{R}$ satisfies

$$t_0\lambda_1 = \frac{v_{1,1}}{v_{1,2}}, \qquad G(\alpha) = \begin{bmatrix} 1 & \alpha \\ 0 & 1 \end{bmatrix}.$$

*Remark* 3.4 (Measure-zero exclusions). The complements of the admissible sets $\mathcal{X}_e$, $\mathcal{X}_h$, and $\mathcal{X}_p$ are algebraic hypersurfaces in $\mathbb{R}^n$ and hence have Lebesgue measure zero. Consequently, these exclusions do not affect practical learning or implementation.

**Notation:** We use $\mathrm{invRep}$ to denote the invariant representation layer in its generic form. When the specific geometric regime of the underlying one-parameter subgroup is relevant, we write $\mathrm{invRep}_e$, $\mathrm{invRep}_h$, and $\mathrm{invRep}_p$ for the elliptic, hyperbolic, and parabolic cases, respectively, as defined in Definitions 3.1, 3.2, and 3.3. Unless otherwise stated, theoretical results are expressed using the unified notation $\mathrm{invRep}$ and apply uniformly across all three regimes.

The significance of this construction is that $\mathrm{invRep}(x)$ provides a unique identifier for each orbit. We first establish that this representation is geometrically consistent with the subgroup action.

**Proposition 3.5** (Orbit Representation). *Let $G$ be a one-parameter subgroup of type E, H, or P, with invariant representation $\mathrm{invRep}(\cdot; A, \lambda)$. Then for any $x \in \mathbb{R}^n$,*

$$\mathcal{O}_G(x) = \mathcal{O}_G\big(A^{-1} \mathrm{invRep}(x; A, \lambda)\big). \qquad (15)$$

Proposition 3.5 ensures that the representation mapping does not leave the group manifold. Beyond consistency, we must guarantee that the representation is *complete*, meaning it does not lose information by collapsing distinct orbits.

**Proposition 3.6** (Orbit Separation). *The mapping $\mathrm{invRep}$ induces a bijection between the orbit space $\mathcal{X}/G$ and its image. Specifically, for any admissible $x, x'$,*

$$\mathcal{O}_G(x) = \mathcal{O}_G(x') \iff \mathrm{invRep}(x) = \mathrm{invRep}(x').$$

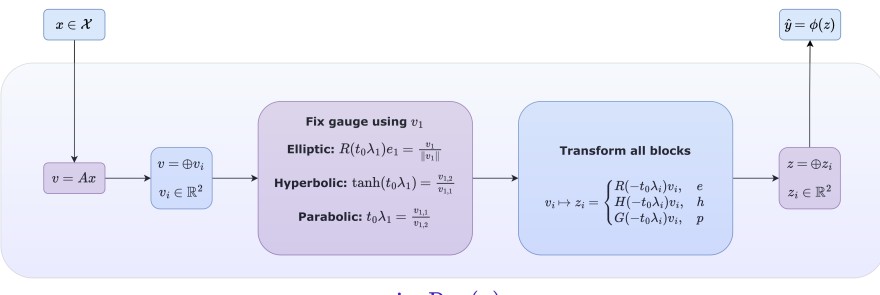

invRep($x$)

*Figure 2.* **The $H_\gamma$-Net Architectural Pipeline for Joint Symmetry Discovery and Invariant Learning.** The input $x \in \mathbb{R}^n$ is first projected into a learnable coordinate system $v = Ax$. The framework then performs a *gauge fixing* step using the first 2D block $v_1$ to solve for the canonicalization parameter $t_0$ across three geometric regimes: Elliptic (rotation), Hyperbolic (boost), and Parabolic (shear). This parameter is applied to transform the remaining blocks into a fixed canonical orientation, effectively mapping the input to an orbit-separating representation $z = \text{invRep}(x)$. Finally, a network $\phi$ operates on $z$ to produce the prediction $\hat{y}$, ensuring structural invariance to the discovered one-parameter subgroup. For clarity, only the principal hyperbolic construction is shown; see Definition 3.2 and related invariant definitions for full details.

Propositions 3.5 and 3.6 together imply that $\text{invRep}$ is strictly $H_\gamma$-invariant and orbit-separating, and therefore constitutes a complete invariant of the one-parameter subgroup action.

Combining orbit invariance and separation leads to our central theoretical result, which defines the necessary and sufficient functional form for any model respecting 1D subgroup symmetry.

**Theorem 3.7** (Canonical Form for Invariant Functions). *Let $H_\gamma$ be a one-parameter subgroup of type E, H, or P. A function $\Psi : \mathcal{X} \to \mathbb{R}^m$ is G-invariant if and only if it can be written as*

$$\Psi(x) = \phi(\text{invRep}(x)), \qquad (16)$$

*where $\phi : \mathbb{R}^n \to \mathbb{R}^m$ is an unconstrained function.*

Proofs of all stated theoretical results are provided in Appendix C.

### 3.3. Symmetry Discovery Framework

Theorem 3.7 provides the architectural foundation of our symmetry discovery framework. By restricting the predictive model to the form

$$f_\theta = \phi_w \circ \text{invRep},$$

where $\phi_w$ is a neural network (typically an MLP), we obtain a hypothesis class that is *structurally guaranteed* to be invariant under a one-parameter subgroup. An overview of the $H_\gamma$-Net pipeline is shown in Figure 2, illustrating the canonicalization of inputs via $\text{invRep}$ and its integration with the prediction network.

To enable automatic symmetry discovery, we make the dependence of the invariant representation on the subgroup

parameters explicit during optimization. Specifically, the subgroup is parameterized by $\theta_{\text{sym}} = \{A, \lambda\}$, which are treated as learnable variables. The model is trained to jointly discover the subgroup and learn the task-specific mapping by minimizing the empirical risk

$$\min_{w, A, \lambda} \mathcal{L} = \frac{1}{N} \sum_{(x,y) \in \mathcal{D}} \left\| \phi_w(\text{invRep}(x; A, \lambda)) - y \right\|^2 \quad (17)$$

This framework offers significant advantages over post-training methods: the discovery process actively informs the functional learning, providing a strong inductive bias that improves data efficiency. Furthermore, the model is fully interpretable; once trained, the parameters $A$ and $\lambda$ explicitly define the recovered generator $B = A^{-1} L_0 A$, allowing for the direct identification of physical symmetries such as rotations, relativistic boosts, or shears from raw data.

## 4. Experiments

We evaluate $H_\gamma$-Net on synthetic and real-world tasks involving functions invariant to one-parameter subgroups, with the goal of jointly discovering the underlying symmetry $H_\gamma$ and learning functions invariant to the discovered subgroup. We compare against Augerino (Benton et al., 2020), a representative method for symmetry discovery via sampling-based augmentation. All experiments follow the architectural pipeline in Figure 2 for our method, employing regime-specific invariant representations with identical prediction backbones across settings.

**Baselines.** We compare against Augerino (Benton et al., 2020), the only baseline, to the best of our knowledge, that jointly performs symmetry discovery and function learning for continuous symmetries. Other related approaches, including LieGAN (Yang et al., 2023) and Bispectral Neural

Networks (BNNs) (Sanborn et al., 2022), are not included as primary baselines; their scope and applicability relative to our setting are summarized in Table 1.

We evaluate each method using three key metrics:

(i) **Prediction error (MSE) or accuracy**, measuring prediction error or accuracy;

(ii) **Cosine Similarity**, quantifying how well the learned generator aligns with the target Lie algebra: we project the learned generator onto the reference Lie algebra and compute the cosine similarity between the learned generator and its projection.

(iii) **Invariance Error**, defined as $\mathbb{E}_{h,\,x}\left[\|f(x) - f(h \cdot x)\|^2\right]$, which measures consistency of true function $f$ under transformations $h$ from a given subgroup $H$.

### 4.1. Synthetic Tasks: Learning $H_\gamma$-Invariant Functions

We first evaluate $H_\gamma$-Net on regression tasks where the symmetries are unknown and must be recovered from data.

**Invariant Polynomial Regression.** We consider regression tasks defined by polynomial functions that are exactly invariant under a hidden one-parameter subgroup of $SO(n)$. The target function is of the form

$$u(x) = w^\top \operatorname{vec}\left( \sum_{i=1}^{n/2} b_i\, p_i(v_i v_i^\top) \right),$$

where $v$ is as defined in equation 7, the weight vector $w$ is chosen such that $f$ is invariant under the group action $x \mapsto A^\top \bigoplus_i R(\theta_i)\, A\, x$, with unknown $A \in SO(n)$ and planar rotations $R(\theta_i)$. This task evaluates the ability of the model to jointly learn the polynomial mapping and recover the underlying rotational generator.

**Angled Sine Function.** To test non-polynomial periodic structure under the discovered symmetry, we define the *Angled Sine* regression task. Let $v$ and its 2D block decomposition $\{v_i\}$ be as in equation 7. For each block $v_i = (v_{i,x}, v_{i,y})$, define $r_i = \|v_i\|_2$ and $\theta_i = \operatorname{atan2}(v_{i,y}, v_{i,x})$. The target function is

$$g(x) = r_1 + \sum_{j \geq 2} r_j\, \sin(\theta_j - \lambda_j \theta_1), \qquad (18)$$

where $\{\lambda_j\}$ are fixed phase-coupling (rotation-rate) coefficients. This construction is inspired by Kuramoto-type phase coupling (Kuramoto, 1975; Strogatz, 2000), and is deliberately sensitive to generator errors: small misalignment of the recovered subgroup basis induces phase drift and degrades prediction, thereby providing a stringent test of symmetry discovery beyond polynomial invariants.

### 4.2. Real-World Symmetry Discovery

**Double Pendulum with Spring Coupling.** We predict the spring force $f = \pm k(q_1 - q_2)$ from the positions and momenta of two coupled pendulums. This system possesses a symmetry in the configuration space corresponding to the diagonal subgroup $\Delta(SO(2)) \subset SO(2) \times SO(2)$, characterizing a simultaneous rotation of both pendulums. The momenta, however, remain non-invariant under this transformation. $H_\gamma$-Net must discover this partial symmetry to effectively generalize.

**Top Quark Tagging.** We apply our framework to the classification of high-energy jet physics data (Bogatskiy et al., 2020; Komiske et al., 2019). In the *Top Tagging* task, the goal is to distinguish top quark jets from background QCD jets based on jet constituents' four-momenta. The underlying physical laws are invariant under Lorentz transformations, including boosts and rotations. $H_\gamma$-Net is tasked with discovering the specific one-parameter boost or rotation subgroup that characterizes the jet distribution to improve tagging robustness.

Additional experiments, covering moment of inertia prediction (G.2), parabolic symmetry discovery (G.1), dataset size scaling (Figure 4a), robustness to noise (Figure 4b), and extended baseline comparisons (G.3, G.4), are presented in Appendix G.

## 5. Results

Tables 2 and 3 summarize the empirical performance of $H_\gamma$-Net compared to Augerino across various tasks. Overall, $H_\gamma$-Net consistently achieves lower prediction error, substantially reduced invariance error, and near-perfect recovery of the underlying Lie algebra generator, highlighting the benefits of enforcing symmetry by construction.

### 5.1. Synthetic regression tasks.

On all regression benchmarks in Table 2, $H_\gamma$-Net outperforms Augerino across all metrics. The improvements are particularly pronounced in invariance error, where $H_\gamma$-Net achieves reductions of several orders of magnitude. This reflects the fact that invariance in $H_\gamma$-Net is guaranteed structurally via the invariant representation layer, whereas Augerino only approximates invariance through learned augmentation distributions.

Generator recovery, measured by cosine similarity, further illustrates this distinction. While Augerino exhibits noticeable variance and degraded alignment, especially on the Angled Sine task, $H_\gamma$-Net consistently recovers the true generator with cosine similarity close to $1.0$. The Angled Sine task is especially sensitive to generator misalignment due to phase coupling across blocks, and the strong per-

*Table 2.* Comparison of Augerino and $H_\gamma$-Net (Elliptical) on Regression Tasks. Prediction error and invariance error are reported as (mean $\pm$ std) $\times 10^k$. Lower is better ($\downarrow$), higher is better ($\uparrow$).

| Task | Metric | Augerino | $H_\gamma$-Net (e) |
|---|---|---|---|
| **Double Pendulum** | Prediction Error (MSE $\downarrow$) | $(6.2 \pm 2.9) \times 10^{-4}$ | $(\mathbf{2.0} \pm 0.1) \times 10^{-4}$ |
| | Cosine Similarity ($\uparrow$) | $0.908 \pm 0.102$ | $\mathbf{1.000} \pm 0.000$ |
| | Invariance Error ($\downarrow$) | $(4.7 \pm 4.8) \times 10^{-4}$ | $(\mathbf{0.9} \pm 0.4) \times 10^{-5}$ |
| **Invariant Polynomial u(x)** | Prediction Error (MSE $\downarrow$) | $(1.7 \pm 1.4) \times 10^{0}$ | $(\mathbf{1.0} \pm 0.0) \times 10^{-5}$ |
| | Cosine Similarity ($\uparrow$) | $0.998 \pm 0.000$ | $\mathbf{0.999} \pm 0.000$ |
| | Invariance Error ($\downarrow$) | $(7.8 \pm 4.3) \times 10^{-2}$ | $(\mathbf{0.4} \pm 0.2) \times 10^{-5}$ |
| **Angled Sine** | Prediction Error (MSE $\downarrow$) | $(3.4 \pm 4.7) \times 10^{-2}$ | $(\mathbf{1.1} \pm 1.2) \times 10^{-2}$ |
| | Cosine Similarity ($\uparrow$) | $0.616 \pm 0.075$ | $\mathbf{1.000} \pm 0.000$ |
| | Invariance Error ($\downarrow$) | $(9.2 \pm 14.0) \times 10^{-3}$ | $(\mathbf{1.6} \pm 2.1) \times 10^{-3}$ |

*Table 3.* Top-quark tagging performance under different symmetry discovery regimes, where $H_\gamma$-Net (Elliptic) and $H_\gamma$-Net (Hyperbolic) employ $\mathrm{invRep}_e$ and $\mathrm{invRep}_h$ to discover rotational and boost generators of the Lorentz group, respectively; invariance error is unavailable due to the absence of ground-truth targets.

| Model | Accuracy ($\uparrow$) | Cosine Similarity ($\uparrow$) |
|---|---|---|
| Augerino | $74.9 \pm 2.4\%$ | $0.994 \pm 0.003$ |
| $H_\gamma$-Net (e) | $83.0 \pm 0.1\%$ | $1.000 \pm 0.000$ |
| $H_\gamma$-Net (h) | $84.4 \pm 0.4\%$ | $0.996 \pm 0.003$ |

formance of $H_\gamma$-Net confirms that the learned subgroup parameters accurately capture the underlying one-parameter symmetry rather than merely improving prediction error.

## 5.2. Real-world symmetry discovery.

Table 3 reports results on the Top Quark Tagging task, where the relevant symmetry arises from the Lorentz group. Both elliptic and hyperbolic variants of $H_\gamma$-Net significantly outperform Augerino in classification accuracy while maintaining near-perfect generator recovery. Notably, $H_\gamma$-Net (Hyperbolic) achieves the highest accuracy, consistent with the presence of boost-type symmetries in high-energy jet data. These results demonstrate that the framework can successfully identify and exploit non-compact symmetries in realistic, high-dimensional settings.

## 6. Discussion

We highlight the important implications of $H_\gamma$-Net for symmetry discovery and invariant learning; detailed explanations are deferred to Appendix H.

### 6.1. Richness of elliptic orbits

In the elliptic regime, a one-parameter action generally operates across multiple 2D planes with distinct rates $\{\lambda_i\}$, producing closed but typically non-circular orbits (purely circular motion arises only in the single-plane degenerate case).

The proposed $\mathrm{invRep}$ reflects this geometry via a $t_0$-based canonicalization: unlike blockwise norm invariants, which can inadvertently become invariant to larger subgroups and fail to separate one-parameter orbits for $n \geq 4$, our construction is designed to be invariant and orbit-separating across elliptic, hyperbolic, and parabolic regimes.

### 6.2. Avoiding the symmetry-free solution

A common failure mode in symmetry discovery is convergence to a symmetry-free predictor (i.e., effectively learning the identity transformation). $H_\gamma$-Net rules this out by construction: constraining the hypothesis class to $f = \phi \circ \mathrm{invRep}$ ensures every admissible predictor is structurally invariant to some one-parameter subgroup $H_\gamma$, so a symmetry-free solution is not representable.

### 6.3. Higher-dimensional symmetry discovery

Although $H_\gamma$-Net is designed for one-parameter subgroups, it can be used as a primitive for recovering higher-dimensional Lie algebras. We consider two generic extensions:

(i) *parallel discovery*, where multiple independent random initializations recover different one-dimensional symmetry directions, and

(ii) *sequential discovery*, where new generators are learned with orthogonality penalties against previously recovered ones, and additional directions are obtained by Lie brackets. A detailed description of these procedures is provided in Appendix H.1. Table 4 summarizes the resulting higher-dimensional recovery experiment on Top Quark Tagging, where the relevant rotational subalgebra is contained in the Lorentz algebra.

These results show that the discovered generators are not merely isolated one-dimensional symmetries: their span closely matches the target rotational subalgebra, with high generator-level alignment and small subspace-level errors. The cosine-regularized strategy further decorrelates the

*Table 4.* Higher-dimensional symmetry discovery on Top Quark Tagging. We evaluate recovery of the rotational subalgebra using two compositional strategies: independent random initializations and sequential discovery with cosine regularization plus Lie-bracket closure. Cosines report alignment of recovered generators with an orthonormal basis of the target subalgebra. Principal angles, projection distance, and Grassmann distance measure subspace-level recovery. Lower is better for angles and distances.

| Setting | $\cos_1 \uparrow$ | $\cos_2 \uparrow$ | $\cos_3 \uparrow$ | Principal angles $\downarrow$ | Proj. dist. $\downarrow$ | Grass. dist. $\downarrow$ |
|---|---|---|---|---|---|---|
| Random inits | 0.9910 | 0.9962 | 0.9985 | $(1.74°, 8.82°, 9.43°)$ | 0.2264 | 0.2274 |
| Cos-reg + bracket | 0.9910 | 0.9972 | 0.9896 | $(0.52°, 8.24°, 8.84°)$ | 0.2103 | 0.2110 |

learned directions, and Lie-bracket closure yields a coherent multi-dimensional algebraic structure.

### 6.4. Separably mixed generators

The elliptic, hyperbolic, and parabolic regimes should be viewed as primary geometric building blocks, not as a complete partition of all one-parameter generators. In higher dimensions, a generator may decompose into independent invariant subspaces carrying different block types; we call these *separably mixed* generators. The invariant representation extends naturally to this case: choose an admissible pivot block, estimate the common orbit parameter $t_0$ using the inverse alignment rule for the pivot type, and apply the corresponding closed-form inverse transformation to every other block according to its own type. Thus, all blocks share the same global group parameter, while each block uses its own elliptic, hyperbolic, or parabolic transformation. This preserves the same canonicalization principle and the same invariance/orbit-separation guarantees. Full definitions and proofs are provided in Appendix D.

This extension also clarifies the scope of the framework. Together, the pure and separably mixed cases cover the regular block-decomposable part of the generator space, where the generator admits a decomposition into independent canonical components. The excluded cases are inseparably mixed or degenerate generators, where different components are coupled through defective eigenspaces or singular Jordan structure. Such degeneracies are characterized by algebraic constraints on the generator parameters, such as vanishing discriminants or loss of a complete eigenbasis. They therefore lie in a lower-dimensional algebraic subset of the ambient generator space, and hence have Lebesgue measure zero. Thus, in a probabilistic sense, the excluded cases occur with probability zero under any absolutely continuous distribution over generator parameters, and the proposed framework covers almost all generators.

### 6.5. Multi-pivot canonicalization

In our experiments, we did not observe practical optimization issues from the local non-smoothness of canonicalization. Nevertheless, a fixed pivot block can in principle become ill-conditioned near the boundary of its admissible domain. This potential sensitivity can be reduced by using multiple candidate pivots: each pivot defines an invariant representation, and the resulting charts can be concatenated or smoothly combined using invariant confidence weights that favor well-conditioned pivots. Since both the charts and the weights are invariant, the multi-pivot representation remains invariant while reducing dependence on any single pivot. Details are provided in Appendix H.2.

## 7. Limitations

Our framework assumes that the geometric regime of the underlying one-parameter subgroup, elliptic, hyperbolic, or parabolic, is specified *a priori*. Jointly inferring the regime and subgroup parameters is fundamentally challenging due to trade-offs between generality, interpretability, and identifiability. Admitting all regimes simultaneously would require a hypothesis space mixing compact, semisimple, and nilpotent generators, preventing closed-form, orbit-separating invariant constructions and destabilizing joint symmetry discovery and function learning. Conditioning on a fixed regime enables provably invariant, interpretable architectures with identifiable Lie algebra parameters.

In the parabolic case, we restrict attention to generators with nilpotency index two, which yields a simple canonical block structure and a well-posed gauge-fixing procedure. Extending the framework to higher-order nilpotent generators is a natural direction for future work, but would require new canonicalization and orbit-normalization mechanisms.

## 8. Conclusion

We introduced a modular framework for joint function learning and automatic discovery of one-parameter symmetries across elliptic, hyperbolic, and parabolic regimes. By embedding symmetry directly into the architecture via regime-specific invariant representations, our models are provably invariant, orbit-separating, and structurally interpretable, while enabling end-to-end recovery of the underlying Lie algebra generator. Experiments on synthetic and real-world tasks demonstrate accurate symmetry discovery and strong predictive performance in both compact and non-compact settings. This work provides a principled foundation for interpretable symmetry discovery in learning systems and opens avenues toward richer continuous symmetry classes.

## Impact Statement

This work aims to improve the reliability and interpretability of machine learning models for scientific and geometric data by enabling neural networks to discover and respect hidden continuous symmetries directly from data. By building the discovered symmetry into the model architecture, the proposed framework can improve data efficiency, robustness, and transparency in applications such as physical simulation, dynamical systems, and high-energy physics.

As with other methods that impose structural assumptions, the approach may perform poorly if the assumed symmetry regime is inappropriate for the data. Care should therefore be taken when applying the method in high-stakes scientific or engineering settings, where incorrect symmetry assumptions could lead to misleading predictions. We do not foresee direct negative societal impacts beyond those common to machine learning methods used for scientific modeling.

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

## A. Notation

Table 5 summarizes the symbols and operators used throughout the paper. Unless otherwise stated, vectors are column vectors and all matrices are real-valued.

| Symbol | Description |
|---|---|
| $n$ | Ambient input dimension |
| $x \in \mathbb{R}^n$ | Input vector |
| $G \subset GL(n)$ | Matrix Lie group acting linearly on $\mathbb{R}^n$ |
| $\mathfrak{g}$ | Lie algebra of $G$ |
| $B \in \mathfrak{g}$ | Lie algebra generator |
| $H_\gamma$ | One-parameter subgroup $\{\exp(tB) \mid t \in \mathbb{R}\}$ |
| $\gamma(t)$ | Group homomorphism $\gamma(t) = \exp(tB)$ |
| $\mathcal{O}_G(x)$ | Orbit of $x$ under the action of $G$ |
| $A \in GL(n)$ | Learnable orientation / change-of-basis matrix |
| $v = Ax$ | Input expressed in subgroup-aligned coordinates |
| $v_i \in \mathbb{R}^2$ | $i$-th two-dimensional block of $v$ |
| $\oplus$ | Direct sum of vectors or matrices |
| $\bigoplus$ | Block-diagonal (external) direct sum over multiple blocks |
| $t_0(x)$ | Canonicalization (gauge-fixing) parameter inferred from $v_1$ |
| $\mathrm{invRep}(x)$ | Invariant, orbit-separating representation of $x$ |
| $\mathrm{invRep}_e$ | Elliptic (rotational) invariant representation |
| $\mathrm{invRep}_h$ | Hyperbolic (boost) invariant representation |
| $\mathrm{invRep}_p$ | Parabolic (shear) invariant representation |
| $\mathcal{X}_e, \mathcal{X}_h, \mathcal{X}_p$ | Admissible domains for each regime |
| $\varphi$ | backbone prediction network |
| $f = \varphi \circ \mathrm{invRep}$ | Canonical form of an invariant function |
| $\Psi_{\mathrm{eq}}$ | Equivariant function obtained via decanonicalization |

*Table 5.* Summary of notation used throughout the paper.

## B. Background

**Definition B.1.** A **group** $(G, \cdot)$ is a set $G$ with a binary operation $\cdot : G \times G \to G$ satisfying associativity, identity, and inverse properties.

**Definition B.2.** A **subgroup** $(H, \cdot)$ of a group $G$ is a subset $H \subseteq G$ that forms a group under the group operation of $G$.

**Definition B.3.** A **group action** of a group $G$ on a set $X$ is a function $\phi : G \times X \to X$ satisfying: $\phi(e, x) = x, \forall x \in X$, and $\phi(g, \phi(h, x)) = \phi(gh, x), \forall g, h \in G, x \in X$.

**Definition B.4.** The **orbit** of an element $x \in X$ under a group action $\phi$ is the set $\mathrm{Orb}(x) = \{\phi(g, x) \mid g \in G\}$.

**Definition B.5.** A **homomorphism** between two groups $G$ and $H$ is a function $\varphi : G \to H$ satisfying $\varphi(g_1 g_2) = \varphi(g_1)\varphi(g_2)$ for all $g_1, g_2 \in G$.

**Definition B.6.** The **special orthogonal group** $SO(n)$ is the group of $n \times n$ real matrices $R$ satisfying $R^T R = I$ and $\det(R) = 1$.

**Definition B.7.** A **2D rotation matrix** is a $2 \times 2$ matrix of the form $R(t) = \begin{bmatrix} \cos t & -\sin t \\ \sin t & \cos t \end{bmatrix}$.

**Definition B.8.** A **3D rotation matrix about the $z$-axis** is given by, $R_z(t) = \begin{bmatrix} R(t) & \mathbf{0} \\ \mathbf{0}^T & 1 \end{bmatrix}$, where $R(t)$ is the 2D rotation matrix as defined in B.7.

# C. Proofs

## C.1. Proof of Proposition 3.5 (Elliptic)

**Proposition 3.5** (Orbit Representation)**.** *Let $G$ be a one-parameter subgroup of type E, H, or P, with invariant representation* $\mathrm{invRep}(\cdot; A, \lambda)$. *Then for any* $x \in \mathbb{R}^n$,

$$\mathcal{O}_G(x) = \mathcal{O}_G\big(A^{-1}\,\mathrm{invRep}(x; A, \lambda)\big). \tag{15}$$

*Proof.* In this case $\mathrm{invRep} = \mathrm{invRep}_e$.

Let $y = A^{-1}\,\mathrm{invRep}_e(x)$. We must show that $y$ lies on the orbit of $x$, i.e., $y \in \mathcal{O}_G(x)$. An element $h_t \in G$ is defined as $h_t = \exp(tA^{-1}L_0 A) = A^{-1}\exp(tL_0)A$. Let $v = Ax$, then the action on $x$ is $h_t x = A^{-1}\exp(tL_0)v$. Since $\exp(tL_0)$ is block-diagonal, we have $h_t x = A^{-1}\bigoplus_{i=1}^{n/2}\hat{h}_t^{(i)}v_i$, where $\hat{h}_t^{(i)}$ is the canonical action (rotation, squeeze, or shear) for the $i$-th block.

By Definition 3.1–3.3, the parameter $t_0$ is chosen such that the transformation $\hat{h}_{-t_0}^{(1)}$ maps $v_1$ to the canonical reference vector ($k\mathbf{e}_1$ or $k\mathbf{e}_2$). Specifically, for all regimes, $\mathrm{invRep}_e(x) = \hat{h}_{-t_0}^{(1)}v_1 \oplus \bigoplus_{i=2}^{n/2}\hat{h}_{-t_0}^{(i)}v_i$. Choosing $t = -t_0$, we obtain:

$$h_{-t_0}x = A^{-1}\left(\hat{h}_{-t_0}^{(1)}v_1 \oplus \bigoplus_{i=2}^{n/2}\hat{h}_{-t_0}^{(i)}v_i\right) = A^{-1}\,\mathrm{invRep}_e(x) = y.$$

Since there exists $t = -t_0$ such that $h_t x = y$, it follows that $y \in \mathcal{O}_G(x)$, and consequently $\mathcal{O}_G(x) = \mathcal{O}_G(y)$. $\qquad\square$

## C.2. Proof of Proposition 3.6 (Elliptic)

**Proposition 3.6** (Orbit Separation)**.** *The mapping* $\mathrm{invRep}$ *induces a bijection between the orbit space* $\mathcal{X}/G$ *and its image. Specifically, for any admissible* $x, x'$,

$$\mathcal{O}_G(x) = \mathcal{O}_G(x') \iff \mathrm{invRep}(x) = \mathrm{invRep}(x').$$

*Proof.* In this case $\mathrm{invRep} = \mathrm{invRep}_e$ and $A^{-1} = A^{-1}$.

($\Longleftarrow$) Assume $\mathrm{invRep}_e(x) = \mathrm{invRep}_e(x')$. By Proposition 3.5, $\mathcal{O}_G(x) = \mathcal{O}_G(A^{-1}\,\mathrm{invRep}_e(x)) = \mathcal{O}_G(A^{-1}\,\mathrm{invRep}_e(x')) = \mathcal{O}_G(x')$.

($\Longrightarrow$) Assume $\mathcal{O}_G(x) = \mathcal{O}_G(x')$, meaning $x' = h_t x$ for some $t \in \mathbb{R}$. We show $\mathrm{invRep}_e(h_t x) = \mathrm{invRep}_e(x)$. Let $v = Ax$ and $v(t) = A(h_t x) = \exp(tL_0)v$. The components are $v_i(t) = \hat{h}_t^{(i)}v_i$. For all regimes, the invariant $N(v_1)$ is preserved: $N(v_1(t)) = N(v_1)$. The new canonical parameter $t_0'$ for $v(t)$ must satisfy $\hat{h}_{t_0'}^{(1)}\mathbf{e}_{ref} = v_1(t)/N(v_1)$. Since $v_1 = N(v_1)\hat{h}_{t_0}^{(1)}\mathbf{e}_{ref}$, we have $\hat{h}_{t_0'}^{(1)}\mathbf{e}_{ref} = \hat{h}_t^{(1)}\hat{h}_{t_0}^{(1)}\mathbf{e}_{ref} = \hat{h}_{t+t_0}^{(1)}\mathbf{e}_{ref}$, which implies $t_0' = t + t_0$. Constructing $\mathrm{invRep}_e(h_t x)$:

$$\mathrm{invRep}_e(h_t x) = \hat{h}_{-(t+t_0)}^{(1)}v_1(t) \oplus \bigoplus_{i=2}^{n/2}\hat{h}_{-(t+t_0)}^{(i)}v_i(t)$$

$$= \hat{h}_{-t-t_0}^{(1)}\left(\hat{h}_t^{(1)}v_1\right) \oplus \bigoplus_{i=2}^{n/2}\hat{h}_{-t-t_0}^{(i)}\left(\hat{h}_t^{(i)}v_i\right)$$

$$= \hat{h}_{-t_0}^{(1)}v_1 \oplus \bigoplus_{i=2}^{n/2}\hat{h}_{-t_0}^{(i)}v_i = \mathrm{invRep}_e(x).$$

Thus, $\mathrm{invRep}_e$ is a complete invariant of the action of $G$. $\qquad\square$

**Remark (Role of $A$ in the elliptic case).** Let $L_0 \in \mathfrak{so}(n)$ be a fixed generator and let $A \in GL(n)$ be invertible. The one-parameter subgroup

$$G_A = \left\{A^{-1}\exp(tL_0)A : t \in \mathbb{R}\right\} \subset GL(n)$$

is conjugate to $\{\exp(tL_0)\}_{t\in\mathbb{R}}$ and defines the group action used by our elliptic construction. All invariance statements that follow from canonicalization/orbit-representation apply to this conjugated subgroup for any invertible $A$.

When domain knowledge indicates that the underlying elliptic symmetry is a one-parameter subgroup of $SO(n)$, we additionally encourage $A$ to be (approximately) orthogonal in training by adding the regularizer

$$\lambda_A \left\| A^\top A - I \right\|_F^2,$$

so that $A$ stays close to $O(n)$ (and hence $G_A$ stays close to a subgroup of $SO(n)$). Alternatively, one may parameterize $A \in SO(n)$ directly to enforce orthogonality exactly.

### C.3. Proofs for the hyperbolic regime

*Proof of Proposition 3.6 (hyperbolic case).* Let $x \in X_h$ and define $v := Ax$. We consider the case of $|v_{1,1}| > |v_{1,2}|$ and the second case follows similarly. In aligned coordinates, the generator has block-diagonal form

$$L_0 = \bigoplus_{i=1}^{n/2} B_{\mathrm{hyp}}(\lambda_i), \qquad B_{\mathrm{hyp}}(\lambda) = \begin{bmatrix} 0 & \lambda \\ \lambda & 0 \end{bmatrix},$$

so that

$$h_t x = A^{-1} \exp(tL_0)v = A^{-1}\Big( \bigoplus_{i=1}^{n/2} H(t\lambda_i)v_i \Big),$$

where $v_i \in \mathbb{R}^2$ and $H(\cdot)$ denotes the hyperbolic block.

By Definition 3.2, the hyperbolic invariant representation is

$$\mathrm{invRep}_h(x) = \mathrm{sgn}(v_{1,1})N(v_1)e_1 \ \oplus\ \bigoplus_{i=2}^{n/2} H(-t_0\lambda_i)v_i,$$

where $t_0$ is chosen so that

$$H(-t_0\lambda_1)v_1 = \mathrm{sgn}(v_{1,1})N(v_1)e_1.$$

Choosing $t = -t_0$ yields

$$h_{-t_0}x = A^{-1}\Big( \bigoplus_{i=1}^{n/2} H(-t_0\lambda_i)v_i \Big) = A^{-1}\mathrm{invRep}_h(x).$$

Hence $A^{-1}\mathrm{invRep}_h(x)$ lies on the orbit of $x$, and therefore

$$\mathcal{O}_G(x) = \mathcal{O}_G\big(A^{-1}\mathrm{invRep}_h(x)\big).$$

$\square$

*Proof of Proposition 3.7 (hyperbolic case).* ($\Leftarrow$) If $\mathrm{invRep}_h(x) = \mathrm{invRep}_h(x')$, then by Proposition 3.6,

$$\mathcal{O}_G(x) = \mathcal{O}_G\big(A^{-1}\mathrm{invRep}_h(x)\big) = \mathcal{O}_G\big(A^{-1}\mathrm{invRep}_h(x')\big) = \mathcal{O}_G(x').$$

($\Rightarrow$) Assume $\mathcal{O}_G(x) = \mathcal{O}_G(x')$, so $x' = h_t x$ for some $t \in \mathbb{R}$. In aligned coordinates,

$$v(t) := A(h_t x) = \exp(tL_0)v = \bigoplus_{i=1}^{n/2} H(t\lambda_i)v_i.$$

The quantity

$$N(v_i) = \sqrt{|v_{i,1}^2 - v_{i,2}^2|}$$

is invariant under hyperbolic transformations, so $N(v_1(t)) = N(v_1)$. Moreover, on the admissible set $X_h$, $\text{sgn}(v_{1,1})$ is constant along the orbit.

Let $t_0$ be the gauge parameter for $v_1$ and $t_0'$ the corresponding parameter for $v_1(t)$. Using the group property $H(a)H(b) = H(a+b)$, we obtain

$$H(-t_0'\lambda_1)v_1(t) = H(-(t_0'-t)\lambda_1)v_1.$$

By uniqueness of the gauge fixing, this implies $t_0' = t_0 + t$.

Therefore,

$$\text{invRep}_h(h_t x) = \text{sgn}(v_{1,1})N(v_1)e_1 \ \oplus\ \bigoplus_{i=2}^{n/2} H(-(t_0+t)\lambda_i)H(t\lambda_i)v_i = \text{invRep}_h(x).$$

Since $x' = h_t x$, we conclude $\text{invRep}_h(x') = \text{invRep}_h(x)$. $\qquad\square$

### C.4. Proofs for the parabolic regime

*Proof of Proposition 3.6 (parabolic case).* Let $x \in X_p$ and set $v := Ax$. The aligned generator has the block-diagonal form

$$L_0 = \bigoplus_{i=1}^{n/2} B_{\text{par}}(\lambda_i), \qquad B_{\text{par}}(\lambda) = \begin{bmatrix} 0 & \lambda \\ 0 & 0 \end{bmatrix},$$

so that

$$h_t x = A^{-1}\left(\bigoplus_{i=1}^{n/2} S(t\lambda_i)v_i\right), \qquad S(\alpha) = \begin{bmatrix} 1 & \alpha \\ 0 & 1 \end{bmatrix}.$$

By Definition 3.3, the gauge parameter $t_0$ is defined by

$$t_0\lambda_1 = \frac{v_{1,1}}{v_{1,2}},$$

which ensures

$$S(-t_0\lambda_1)v_1 = v_{1,2}e_2.$$

The invariant representation is

$$\text{invRep}_p(x) = v_{1,2}e_2 \ \oplus\ \bigoplus_{i=2}^{n/2} S(-t_0\lambda_i)v_i.$$

Choosing $t = -t_0$ gives

$$h_{-t_0}x = A^{-1}\left(\bigoplus_{i=1}^{n/2} S(-t_0\lambda_i)v_i\right) = A^{-1}\text{invRep}_p(x),$$

so $A^{-1}\text{invRep}_p(x)$ lies on the orbit of $x$, and the orbits coincide. $\qquad\square$

*Proof of Proposition 3.7 (parabolic case).* ($\Leftarrow$) If $\text{invRep}_p(x) = \text{invRep}_p(x')$, then Proposition 3.6 implies $\mathcal{O}_G(x) = \mathcal{O}_G(x')$.

($\Rightarrow$) Assume $x' = h_t x$. In aligned coordinates,

$$v(t) = \bigoplus_{i=1}^{n/2} S(t\lambda_i)v_i.$$

The shear matrix $S(\alpha)$ preserves the second coordinate, hence $v_{1,2}(t) = v_{1,2}$.

Let $t_0$ and $t_0'$ denote the gauge parameters associated with $v_1$ and $v_1(t)$ respectively. Since

$$v_{1,1}(t) = v_{1,1} + t\lambda_1 v_{1,2},$$

we obtain

$$t_0' \lambda_1 = \frac{v_{1,1}(t)}{v_{1,2}} = \frac{v_{1,1}}{v_{1,2}} + t\lambda_1, \quad \text{hence} \quad t_0' = t_0 + t.$$

Using the group law $S(a)S(b) = S(a+b)$, we compute

$$\text{invRep}_p(h_t x) = v_{1,2} e_2 \ \oplus \ \bigoplus_{i=2}^{n/2} S(-(t_0 + t)\lambda_i) S(t\lambda_i) v_i = \text{invRep}_p(x).$$

Therefore $\text{invRep}_p(x') = \text{invRep}_p(x)$. $\qquad \square$

### C.5. Proof of Theorem 3.7 (All)

**Theorem 3.7** (Canonical Form for Invariant Functions). *Let $H_\gamma$ be a one-parameter subgroup of type E, H, or P. A function $\Psi : \mathcal{X} \to \mathbb{R}^m$ is G-invariant if and only if it can be written as*

$$\Psi(x) = \phi(\text{invRep}(x)), \tag{16}$$

*where $\phi : \mathbb{R}^n \to \mathbb{R}^m$ is an unconstrained function.*

*Proof.* We must construct the function $\phi$. Let $Y = \text{Im}(\text{invRep})$ be the set of all possible invariant representations. For any $y \in Y$, we define $\phi$ as:

$$\phi(y) := \Psi(A^{-1} y).$$

First, we show consistency. By Proposition 3.6, each $y \in Y$ corresponds to exactly one orbit $\mathcal{O}_G$. By Proposition 3.5, the point $A^{-1} y$ lies on this orbit. Since $\Psi$ is G-invariant, it takes a constant value on this entire orbit. Therefore, $\phi(y)$ maps the unique orbit identifier $y$ to the unique value $\Psi$ takes on that orbit, making $\phi$ well-defined.

Next, we verify the decomposition for an arbitrary $x \in \mathbb{R}^n$:

$$\phi(\text{invRep}(x)) = \Psi(A^{-1} \text{invRep}(x)).$$

From Proposition 3.5, $x$ and $A^{-1} \text{invRep}(x)$ belong to the same orbit $\mathcal{O}_G(x)$. Since $\Psi$ is G-invariant:

$$\Psi(A^{-1} \text{invRep}(x)) = \Psi(x).$$

Therefore, $\phi(\text{invRep}(x)) = \Psi(x)$, completing the proof. $\qquad \square$

## D. Separably mixed generators.

The elliptic, hyperbolic, and parabolic regimes above should be understood as primary geometric building blocks rather than as a complete partition of all one-parameter generators. In higher dimensions, different invariant subspaces of the same one-parameter action may carry different geometric components. We call such generators *separably mixed* when, after a change of basis, the generator decomposes as a direct sum of independent elliptic, hyperbolic, and parabolic blocks.

Formally, let $H = \{\exp(tB) : t \in \mathbb{R}\}$ act linearly on $\mathbb{R}^n$. We say that $B$ is separably mixed if there exists an invertible matrix $A$ such that

$$B = A^{-1} \widehat{B} A, \qquad \widehat{B} = \widehat{B}^{(1)} \oplus \cdots \oplus \widehat{B}^{(q)},$$

where each block $\widehat{B}^{(r)}$ is of elliptic, hyperbolic, or parabolic type. Consequently,

$$\exp(t\widehat{B}) = \exp(t\widehat{B}^{(1)}) \oplus \cdots \oplus \exp(t\widehat{B}^{(q)}).$$

The pure elliptic, hyperbolic, and parabolic cases correspond to the special case where all nontrivial blocks have the same geometric type. Separably mixed generators therefore allow combinations such as rotation–boost, rotation–shear, or boost–shear actions, while retaining a factorized one-parameter structure.

*Consequently, in the pure regimes, discovering a one-parameter subgroup reduces to learning the orientation $A$ and the canonical generator parameters $\{\lambda_k\}$ under the chosen geometric type. For separably mixed generators, the same principle applies blockwise: the generator is represented as a direct sum of elliptic, hyperbolic, and parabolic components, all driven by the same global subgroup parameter $t$.*

**Invariant representation for separably mixed generators.** The invariant representations provided in the definitions 3.1- 3.3 extend directly to separably mixed generators. Let

$$\widehat{B} = \widehat{B}^{(1)} \oplus \cdots \oplus \widehat{B}^{(q)}$$

be a separably mixed canonical generator, where each block is elliptic, hyperbolic, or parabolic. For $v = Ax$, write

$$v = v^{(1)} \oplus \cdots \oplus v^{(q)}$$

according to the same block decomposition. Choose an admissible pivot block $p$ from one of the components, and compute the corresponding alignment parameter $t_p(x)$ using the appropriate pure-regime rule: elliptic alignment for a rotational pivot, hyperbolic alignment for a boost/squeeze pivot, and parabolic alignment for a shear pivot. We then define

$$\mathrm{invRep}_{\mathrm{mix}}^{(p)}(x) = \exp(-t_p(x)\widehat{B})Ax.$$

Equivalently,

$$\mathrm{invRep}_{\mathrm{mix}}^{(p)}(x) = \bigoplus_{r=1}^{q} \exp(-t_p(x)\widehat{B}^{(r)})v^{(r)}.$$

Thus, the same global alignment parameter $t_p(x)$ is applied to all blocks, while the transformation applied to each block is determined by that block's geometric type. This is the natural mixed analogue of the definitions 3.1- 3.3.

### D.1. Proofs for Separably Mixed Generators

We prove that the arguments used for the pure elliptic, hyperbolic, and parabolic cases extend to separably mixed generators.

Let

$$\widehat{B} = \widehat{B}^{(1)} \oplus \cdots \oplus \widehat{B}^{(q)}$$

be a separably mixed canonical generator, where each $\widehat{B}^{(r)}$ is elliptic, hyperbolic, or parabolic. Let $B = A^{-1}\widehat{B}A$ and $v = Ax$. For an admissible pivot block $p$, define

$$\mathrm{invRep}_{\mathrm{mix}}^{(p)}(x) = \exp(-t_p(x)\widehat{B})Ax.$$

**Invariance.** Let $x' = \exp(sB)x$ for some $s \in \mathbb{R}$. Then, in aligned coordinates,

$$Ax' = A\exp(sB)x = \exp(s\widehat{B})Ax.$$

Since all blocks are driven by the same global subgroup parameter, the pivot alignment satisfies

$$t_p(x') = t_p(x) + s$$

on the admissible domain, up to the stabilizer of the pivot action. Therefore,

$$
\begin{aligned}
\mathrm{invRep}_{\mathrm{mix}}^{(p)}(x') &= \exp(-t_p(x')\widehat{B})Ax' \\
&= \exp(-(t_p(x) + s)\widehat{B})\exp(s\widehat{B})Ax \\
&= \exp(-t_p(x)\widehat{B})Ax \\
&= \mathrm{invRep}_{\mathrm{mix}}^{(p)}(x).
\end{aligned}
$$

Thus $\mathrm{invRep}_{\mathrm{mix}}^{(p)}$ is invariant under the one-parameter action.

**Orbit separation.** Suppose

$$\mathrm{invRep}_{\mathrm{mix}}^{(p)}(x) = \mathrm{invRep}_{\mathrm{mix}}^{(p)}(x').$$

Then

$$\exp(-t_p(x)\widehat{B})Ax = \exp(-t_p(x')\widehat{B})Ax'.$$

Multiplying by $\exp(t_p(x')\widehat{B})$ gives

$$Ax' = \exp((t_p(x') - t_p(x))\widehat{B})Ax.$$

Applying $A^{-1}$ yields

$$x' = \exp((t_p(x') - t_p(x))B)x.$$

Hence $x'$ lies on the same orbit as $x$. The converse direction follows from invariance. Therefore,

$$\mathcal{O}_H(x) = \mathcal{O}_H(x') \quad \Longleftrightarrow \quad \text{invRep}^{(p)}_{\text{mix}}(x) = \text{invRep}^{(p)}_{\text{mix}}(x').$$

**Canonical invariant form.** Let $\Psi$ be invariant under $H$. Since $\text{invRep}^{(p)}_{\text{mix}}$ separates orbits on the admissible domain, define $\phi$ on the image of $\text{invRep}^{(p)}_{\text{mix}}$ by

$$\phi(z) = \Psi(x) \quad \text{for any } x \text{ such that} \quad z = \text{invRep}^{(p)}_{\text{mix}}(x).$$

This is well-defined because points with the same mixed invariant representation lie on the same orbit, and $\Psi$ is constant on orbits. Therefore,

$$\Psi(x) = \phi\left(\text{invRep}^{(p)}_{\text{mix}}(x)\right).$$

This proves that the existing orbit-representation, orbit-separation, and canonical-form results extend to separably mixed generators.

## E. Experimental and Training Settings

Across all experiments, models were trained for **50 epochs**. For the *Invariant Polynomial* u(x), *Angled Sine* g(x), and *Double Pendulum* tasks, we used **32K samples**, while the *Top Tagging* task employed **64K samples**.

For the regression-style tasks (u(x), g(x), Double Pendulum), the backbone network $\phi$ was implemented as a multilayer perceptron (MLP) with hidden layer dimensions $(128, 128, 64, 64, 32)$ and ReLU activations. For the Top Tagging classification task, we used a **ResNet-based** backbone architecture for $\phi$ to better capture the higher-dimensional and structured nature of jet features.

Unless otherwise stated, all other architectural components and optimization hyperparameters were kept fixed across experiments to ensure a fair comparison between symmetry discovery regimes.

## F. Extension from Invariant to Equivariant Functions

The invariant representations defined in Definitions 3.1–3.3 are obtained by *canonicalizing* inputs through an explicit group action. In all three regimes (elliptic, hyperbolic, and parabolic), the construction of $\text{invRep}(x)$ internally applies a transformation of the form $g(-t_0(x))$ to map $x$ to a fixed orbit representative.

### F.1. Canonicalization Mechanism

For any admissible input $x \in \mathcal{X}$, the invariant representation satisfies

$$\text{invRep}(x) = g(-t_0(x)) \cdot x, \tag{19}$$

where $t_0(x)$ is the unique parameter that aligns a designated component of $x$ with a canonical reference configuration. The specific form of $g(\cdot)$ and the equation defining $t_0(x)$ depend on the geometric regime, but the canonicalization principle is identical across all cases.

### F.2. Invariant Functions

By Theorem 3.7, any $H_\gamma$-invariant function $\Psi$ admits the canonical form

$$\Psi(x) = \phi(\text{invRep}(x)),$$

for an unconstrained function $\phi$.

### F.3. Equivariant Function Construction

To construct an equivariant function, we must explicitly undo the canonicalization performed by $\mathrm{invRep}$. Since $\mathrm{invRep}(x)$ is obtained by applying $g(-t_0(x))$ to the input, equivariance is restored by applying the inverse transformation $g(t_0(x))$ to the output of $\phi$.

Let $\phi : \mathbb{R}^n \to \mathcal{Y}$ be a function whose output space $\mathcal{Y}$ carries a compatible action of $H_\gamma$. We define the equivariant function

$$\Psi_{\mathrm{eq}}(x) \;=\; g(t_0(x)) \cdot \phi(\mathrm{invRep}(x)). \tag{20}$$

### F.4. Equivariance Guarantee

**Proposition F.1** (Canonical Form for Equivariant Functions)**.** *The function $\Psi_{\mathrm{eq}}$ defined in equation 20 is $H_\gamma$-equivariant, i.e.,*

$$\Psi_{\mathrm{eq}}(g(t) \cdot x) = g(t) \cdot \Psi_{\mathrm{eq}}(x), \qquad \forall\, t \in \mathbb{R}.$$

*Proof.* Let $x' = g(t) \cdot x$. By invariance of the canonical representation,

$$\mathrm{invRep}(x') = \mathrm{invRep}(x), \qquad t_0(x') = t_0(x) - t.$$

Substituting into equation 20,

$$\begin{aligned}
\Psi_{\mathrm{eq}}(x') &= g(t_0(x')) \cdot \phi(\mathrm{invRep}(x')) \\
&= g(t_0(x) - t) \cdot \phi(\mathrm{invRep}(x)) \\
&= g(t) \cdot g(t_0(x)) \cdot \phi(\mathrm{invRep}(x)) \\
&= g(t) \cdot \Psi_{\mathrm{eq}}(x),
\end{aligned}$$

which proves equivariance. $\qquad\square$

### F.5. Interpretation

The invariant representation $\mathrm{invRep}$ removes the group action by applying $g(-t_0)$ to map inputs to canonical orbit representatives. Equivariant function learning follows a *canonicalize–process–restore* principle: canonicalize the input, apply an unconstrained function in the canonical frame, and restore equivariance by applying the inverse group element $g(t_0)$ at the output.

## G. Additional Experiments

### G.1. Discovery of Parabolic Symmetries

We evaluate the ability of the proposed invariant representation to discover hidden continuous symmetries from raw data. We construct a synthetic regression task governed by a hidden rank-1 nilpotent group action and train a neural network equipped with a learnable parabolic invariant layer ($\mathrm{invRep}_p$) as defined in 3.3.

#### G.1.1. DATASET AND UNDERLYING INVARIANCE

The dataset consists of dense matrices $M \in \mathbb{R}^{n \times n}$ (with $n = 4$) and targets $y \in \mathbb{R}$ generated by a function $f(M)$ composed of a volume term and an invariant regularizer:

$$f(M) = \log|\det(M)| + \lambda_g \cdot g(M). \tag{21}$$

The term $\log|\det(M)|$ is naturally invariant to any transformation in $SL(n)$. The term $g(M)$ is constructed to be specifically invariant to the 1-parameter subgroup $G = \{I + tX \mid t \in \mathbb{R}\}$ generated by a fixed rank-1 nilpotent matrix $X = ab^T$ (where $b^T a = 0$, ensuring $X^2 = 0$).

The invariance of $g(M)$ is defined via two fixed probe vectors $u$ and $w$ satisfying specific algebraic relations with the generator $X$:

$$u^T X = 0 \quad \text{and} \quad w^T X = u^T. \tag{22}$$

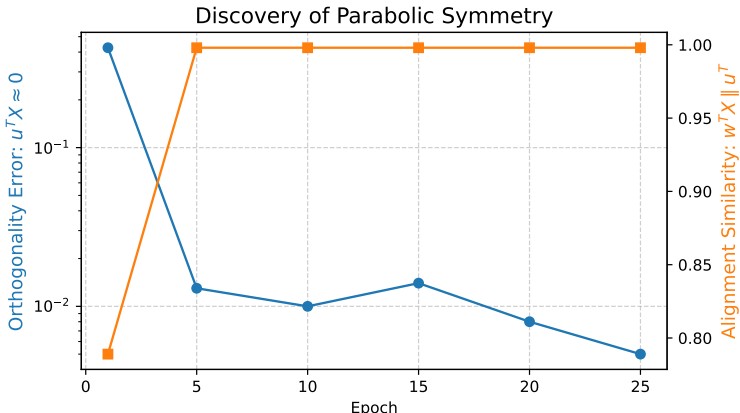

*Figure 3.* **Symmetry Discovery Dynamics.** Starting from a random initialization (Epoch 1), the model rapidly aligns its internal representation with the hidden physical laws of the dataset. Within 5 epochs, the Alignment Similarity (blue) converges to $\approx 1.0$ and Orthogonality Error (red) drops to $\approx 0$, indicating the layer has successfully reconstructed the true generator $X$ from raw data.

*Table 6.* Comparison of Augerino and $H_\gamma$-Net on the Moment of Inertia Prediction Task. Prediction and invariance errors are reported as $(\text{mean} \pm \text{std}) \times 10^k$.

| Metric | Augerino | $H_\gamma$-**Net** |
|---|---|---|
| Prediction Error (MSE $\downarrow$) | $(2.2 \pm 0.5) \times 10^{-4}$ | $(\mathbf{6.6} \pm 0.6) \times 10^{-5}$ |
| Cosine Similarity ($\uparrow$) | $0.985 \pm 0.006$ | $\mathbf{1.000} \pm 0.000$ |
| Invariance Error (True $\downarrow$) | $(7.64 \pm 0.38) \times 10^{-1}$ | $(\mathbf{7.3} \pm 1.0) \times 10^{-3}$ |

The function $g(M)$ computes the difference between two ratios constructed from these probes:

$$g(M) = \log\left(1 + \left(\frac{w^T M v_1}{u^T M v_1} - \frac{w^T M v_2}{u^T M v_2}\right)^2\right), \tag{23}$$

where $v_1, v_2$ are random fixed vectors. Under the group action $M' = (I + tX)M$, the denominator $u^T M' v_i$ remains constant (since $u^T X = 0$), while the numerator shifts linearly by $t(u^T M v_i)$ (since $w^T X = u^T$). Consequently, both ratios shift by exactly $t$, and their difference remains invariant.

### G.1.2. LEARNING DYNAMICS AND GENERATOR RECOVERY

We train a model $\phi(\text{invRep}_p(M))$, where $\phi$ is a standard MLP and $\text{invRep}_p$ is the proposed layer for parabolic subgroups. Crucially, the layer is initialized with a random basis $A$ and random scaling factors $\lambda$. The model must learn the geometry of the hidden generator $X$ solely by minimizing the regression loss.

Any matrix $X_{\text{learned}}$ discovered by the model is considered valid if it lies in the Lie subalgebra defined by the constraints in equation 22. We monitor two geometric metrics during training to verify this discovery:

1. **Orthogonality Error:** Measures if the generator respects the invariant subspace defined by $u$: $\|u^T X_{\text{learned}}\| / \|X_{\text{learned}}\|$.

2. **Alignment Similarity:** Measures if the generator maps the probe $w$ correctly to $u$: cosine-similarity $\left(X_{\text{learned}}^T w, u\right)$.

### G.1.3. RESULTS

As shown in Figure 3, the model exhibits a rapid phase transition in the first 5 epochs. The Alignment Similarity jumps from 0.79 to $> 0.99$, and the Orthogonality Error collapses from 0.43 to $< 0.01$. This confirms that the gradient signal from the downstream task is sufficient to drive the $\text{invRep}_p$ layer to identify the exact 1D subspace of the hidden generator $X$, effectively "locking on" to the underlying physics of the data.

| Method | Cosine Similarity |
|---|---|
| $H_\gamma$-Net (ours) | **1.000000** $\pm$ 0.000000 |
| LieGAN | 0.827722 $\pm$ 0.138458 |
| LieGG | 0.690283 $\pm$ 0.020958 |

*Table 7.* Symmetry-discovery quality on Angled Sine at 32K samples.

### G.2. Moment of Inertia Prediction.

This task requires predicting the inertia matrix $I = \sum_{i=1}^{N} m_i(x_i^T x_i \mathbf{I} - x_i x_i^T)$ from point mass distributions. The inertia tensor is equivariant under the diagonal subgroup $\Delta(SO(3)) = \{(g, \dots, g) \mid g \in SO(3)\} \subset \prod_{i=1}^{N} SO(3)$, reflecting the physical constraint that the same rotation is applied to all particles in the system. While the global symmetry group is three-dimensional, $H_\gamma^{\text{inv}}$-Net is tasked with identifying and recovering a specific one-parameter subgroup (i.e., rotation about a single valid axis) within this subspace. We assess whether the framework can successfully isolate one such valid generator and learn the corresponding equivariant mapping. As shown in Table 6, we see that our method $H_\gamma$-Net outperforms the baseline across all three metrics.

### G.3. Comparison against additional baselines

In the main paper, we use Augerino (Benton et al., 2020) as the primary baseline because it is, to the best of our knowledge, the only method in this comparison set that *jointly* (i) discovers a continuous Lie symmetry and (ii) learns a predictive model for the downstream function mapping.

In this appendix, we therefore provide additional comparisons to some of the baselines: (i) **LieGAN** (Yang et al., 2023), which learns Lie algebra generators but does not produce a supervised prediction model, and (ii) **LieGG** (Moskalev et al., 2022), which performs post-hoc symmetry discovery by extracting a generator from the null space of a Gram-matrix construction.

Table 7 reports a snapshot comparison at a fixed dataset size (32K) under the noise-0 configuration. To further contextualize these results, Fig. 4 (left) compares $H_\gamma$-Net against LieGAN across varying dataset sizes, illustrating the sample efficiency of the proposed method. Fig. 4 (right) analyzes the robustness of $H_\gamma$-Net to increasing noise levels, showing cosine similarity as a function of noise for multiple dataset sizes, with one curve per training regime. Together, these results highlight both improved sample efficiency and graceful degradation under noise for the proposed framework.

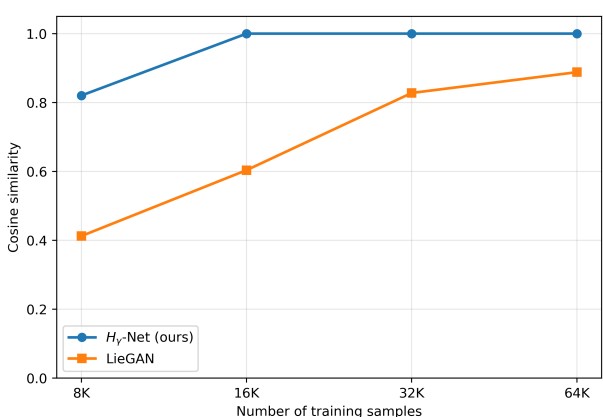 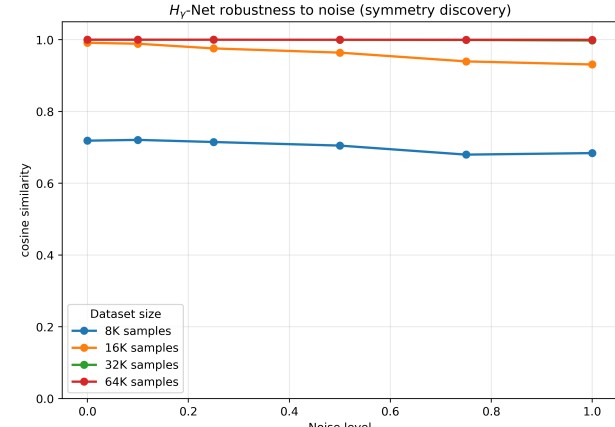

*(a)* Noise-0 configuration: cosine similarity versus number of training samples.

*(b)* $H_\gamma$-Net robustness to noise: cosine similarity versus noise level at various dataset sizes.

*Figure 4.* Additional analysis of symmetry discovery performance on the Angled Sine task. Left: sample efficiency comparison under the noise-0 configuration. Right: robustness of $H_\gamma$-Net to increasing noise across dataset sizes. Only mean values are shown to avoid visual clutter.

## G.4. Comparison with Post-hoc Symmetry Discovery Methods

In the main paper, we use Augerino as the primary baseline because it jointly learns a predictive model and discovers continuous symmetries during training. Here, we provide an additional comparison with two post-hoc symmetry discovery methods, BeyondAffine and LieGG. These methods first fit or use a predictive model and then attempt to extract symmetries from the learned function, for example using Jacobian-based criteria, null-space computations, or polynomial approximations. In contrast, $H_\gamma$-Net incorporates the candidate symmetry directly into the predictive architecture through the invariant representation layer. Thus, the learned predictor is constrained to be invariant/equivariant by construction, while the generator is learned jointly with the downstream task.

We evaluate this distinction on two representative tasks: Top Quark Tagging (TT), a real high-energy physics task with Lorentz/rotational structure, and Double Pendulum (DP), a simulated dynamical system with a rotational invariance. For both tasks, we report the downstream performance and the cosine similarity between the recovered generator and the ground-truth generator. For TT, the performance metric is classification accuracy; for DP, it is mean squared error (MSE). Higher is better for accuracy and cosine similarity, while lower is better for MSE.

*Table 8.* Comparison with post-hoc symmetry discovery methods on Top Quark Tagging (TT) and Double Pendulum (DP). TT reports classification accuracy, while DP reports MSE. Cosine similarity measures alignment between the recovered and ground-truth generators.

| Task | Method | Accuracy / MSE | Cosine similarity |
|---|---|---|---|
| Top Quark Tagging | BeyondAffine | $0.7946 \pm 0.0021$ | $0.5090 \pm 0.0087$ |
| | LieGG | $0.7982 \pm 0.0029$ | $0.9556 \pm 0.0159$ |
| | $H_\gamma$-Net (elliptic) | $0.830 \pm 0.001$ | $\mathbf{1.000 \pm 0.000}$ |
| | $H_\gamma$-Net (hyperbolic) | $\mathbf{0.844 \pm 0.004}$ | $0.996 \pm 0.003$ |
| Double Pendulum | BeyondAffine | $0.01910 \pm 0.00362$ | $0.8703 \pm 0.0803$ |
| | LieGG | $0.01640 \pm 0.00020$ | $0.9848 \pm 0.0118$ |
| | $H_\gamma$-Net (elliptic) | $\mathbf{2.0 \times 10^{-4} \pm 1.0 \times 10^{-5}}$ | $\mathbf{1.000 \pm 0.000}$ |

On Top Quark Tagging, the post-hoc methods achieve accuracies around $0.79$, whereas $H_\gamma$-Net improves the accuracy to $0.830$ in the elliptic setting and $0.844$ in the hyperbolic setting. The difference is also visible in generator recovery. BeyondAffine obtains a substantially lower cosine similarity ($0.5090 \pm 0.0087$), indicating poor alignment with the target generator. LieGG recovers a more accurate generator ($0.9556 \pm 0.0159$), but still remains below the near-perfect recovery of $H_\gamma$-Net. The hyperbolic variant achieves the best task accuracy, while both variants recover the underlying generator with cosine similarity close to one.

On Double Pendulum, the gap in prediction error is more pronounced. BeyondAffine and LieGG obtain MSE values of $0.01910$ and $0.01640$, respectively, while $H_\gamma$-Net reduces the MSE to $2.0 \times 10^{-4}$. This is roughly two orders of magnitude smaller than the post-hoc baselines. The generator recovery follows the same trend: BeyondAffine shows imperfect and higher-variance alignment, LieGG performs better but remains below perfect recovery, and $H_\gamma$-Net recovers the ground-truth generator with cosine similarity $1.000 \pm 0.000$.

These results illustrate the practical limitation of post-hoc symmetry discovery. Since the symmetry is inferred only after fitting, any approximation error in the learned predictor can propagate to the recovered generator. In addition, post-hoc methods often rely on local differential information or null-space estimates, which can be sensitive to optimization error, sampling, or the expressivity of the fitted model. By contrast, $H_\gamma$-Net restricts the hypothesis class during training to functions that are invariant/equivariant with respect to the learned one-parameter subgroup. This couples generator recovery with the downstream objective and yields a predictor whose symmetry is analytically guaranteed by the invariant representation layer. The empirical improvements on both TT and DP therefore support the central design choice of jointly learning the symmetry and the predictive model, rather than discovering symmetry only after training.

# H. Discussion

## H.1. Higher-dimensional symmetry discovery.

Although our main construction targets one-parameter subgroups, it can be used as a modular primitive for discovering higher-dimensional Lie algebras. Let $\mathfrak{h} \subseteq \mathfrak{g}$ denote an unknown $d$-dimensional symmetry algebra. Since every direction $B \in \mathfrak{h}$ generates a one-parameter subgroup $\{\exp(tB) : t \in \mathbb{R}\}$, our one-parameter discovery module can be repeatedly applied to recover a basis or spanning set for $\mathfrak{h}$. We consider two generic strategies.

*Parallel discovery by random initialization.* Run the one-parameter discovery model independently for $M$ random initializations, obtaining candidate generators

$$\widehat{B}_1, \ldots, \widehat{B}_M.$$

When the target symmetry algebra has dimension $d > 1$, different initializations may converge to different one-dimensional symmetry directions. Since any fixed one-dimensional subspace has measure zero inside a higher-dimensional vector space, independent runs are unlikely to recover exactly the same direction unless the optimization landscape strongly favors it. The recovered generators are then normalized and clustered or filtered according to linear independence, for example using cosine similarity or projection distance, to obtain a candidate spanning set

$$\widehat{\mathfrak{h}}_0 = \operatorname{span}\{\widehat{B}_{i_1}, \ldots, \widehat{B}_{i_r}\}.$$

This provides a simple embarrassingly parallel procedure for estimating multiple symmetry directions without modifying the base architecture.

*Sequential discovery with orthogonalization and Lie closure.* Alternatively, generators can be discovered iteratively. Suppose that after $r$ stages we have recovered generators

$$\widehat{B}_1, \ldots, \widehat{B}_r.$$

At stage $r + 1$, we train a new one-parameter model while discouraging collapse onto previously discovered directions using the regularizer

$$\mathcal{L}_{\mathrm{orth}} = \sum_{i=1}^{r} \frac{\langle B, \widehat{B}_i \rangle_F^2}{\|B\|_F^2 \|\widehat{B}_i\|_F^2}.$$

The next generator is obtained by minimizing the task loss together with this decorrelation penalty:

$$\min_{\theta} \ \mathcal{L}_{\mathrm{task}}(\theta) + \beta \mathcal{L}_{\mathrm{orth}}.$$

After recovering several independent directions, we enlarge the candidate algebra by taking Lie brackets

$$[\widehat{B}_i, \widehat{B}_j] = \widehat{B}_i \widehat{B}_j - \widehat{B}_j \widehat{B}_i.$$

The span is then closed iteratively under brackets:

$$\widehat{\mathfrak{h}} = \operatorname{Lie}\big(\widehat{B}_1, \ldots, \widehat{B}_r\big)$$
$$= \operatorname{span}\left\{\widehat{B}_i, [\widehat{B}_i, \widehat{B}_j], [\widehat{B}_i, [\widehat{B}_j, \widehat{B}_k]], \ldots\right\}.$$

In practice, this closure is truncated once newly generated brackets lie within the span of the existing generators, as measured by a projection residual. This gives an algorithmic route to recovering non-abelian symmetry algebras: the one-parameter modules discover local directions, while the bracket operation reconstructs the algebraic structure generated by those directions.

Together, these two procedures extend the proposed framework beyond a single one-parameter subgroup. The random-initialization approach is simple and parallelizable, while the sequential orthogonalization approach gives more direct control over diversity of recovered generators. Lie-bracket closure then converts the discovered one-dimensional directions into a candidate higher-dimensional Lie algebra. Empirically, on Top Quark Tagging, these procedures recover multiple directions in the Lorentz/rotation subalgebra with high alignment, small principal angles, and low Grassmann distance, supporting the view that one-parameter discovery can serve as a scalable building block for higher-dimensional symmetry discovery.

## H.2. Smooth Multi-pivot Canonicalization

In our experiments, we did not observe practical optimization issues arising from the local non-smoothness of the canonicalization map. Nevertheless, canonicalization based on a single pivot block can become numerically ill-conditioned near the boundary of its admissible domain. For example, in the elliptic case, the pivot alignment is ill-conditioned when the pivot norm is close to zero; in the hyperbolic case, instability can occur near the light-cone boundary; and in the parabolic case, the alignment becomes unstable when the denominator used to estimate the shear parameter is close to zero. These singular sets are measure-zero, but samples lying close to them may still lead to numerical sensitivity. A simple mitigation is to replace the single-pivot construction by a multi-pivot, atlas-like construction.

Let $z_j(x) = \mathrm{invRep}^{(j)}(x)$ denote the invariant representation obtained by using block $j$ as the pivot. One direct approach is to concatenate several pivot-based representations:

$$z_{\mathrm{cat}}(x) = \big(z_1(x), z_2(x), \ldots, z_m(x)\big).$$

Since each $z_j$ is invariant on its admissible chart, the concatenated representation is also invariant on the intersection of the corresponding admissible domains:

$$z_{\mathrm{cat}}(g \cdot x) = \big(z_1(g \cdot x), \ldots, z_m(g \cdot x)\big) = \big(z_1(x), \ldots, z_m(x)\big) = z_{\mathrm{cat}}(x).$$

This reduces dependence on a single pivot, but increases the input dimension of the downstream network.

A more compact alternative is to use a smooth weighted combination of pivot charts. Let $s_j(x)$ be a smooth invariant confidence score measuring how well-conditioned pivot $j$ is. Typical choices are

$$s_j^e(x) = \|v_j\|_2^2, \qquad s_j^h(x) = |v_{j,1}^2 - v_{j,2}^2|, \qquad s_j^p(x) = v_{j,2}^2 + \varepsilon,$$

for the elliptic, hyperbolic, and parabolic cases, respectively. Here $v = Ax$, $v_j$ is the $j$-th two-dimensional block, and $\varepsilon > 0$ is a small stabilization constant. Since these scores are invariant under the corresponding one-parameter action, they can be used to define invariant softmax weights

$$\alpha_j(x) = \frac{\exp(s_j(x)/\tau)}{\sum_{\ell=1}^m \exp(s_\ell(x)/\tau)}, \qquad \tau > 0.$$

The smooth multi-pivot representation is then defined as

$$z_{\mathrm{soft}}(x) = \sum_{j=1}^m \alpha_j(x) z_j(x).$$

This representation remains invariant wherever the selected pivot charts are defined. Indeed, since each $z_j$ is invariant and each confidence score $s_j$ is invariant, we have

$$\alpha_j(g \cdot x) = \alpha_j(x), \qquad z_j(g \cdot x) = z_j(x).$$

Therefore,

$$
\begin{aligned}
z_{\mathrm{soft}}(g \cdot x) &= \sum_{j=1}^m \alpha_j(g \cdot x) z_j(g \cdot x) \\
&= \sum_{j=1}^m \alpha_j(x) z_j(x) \\
&= z_{\mathrm{soft}}(x).
\end{aligned}
$$

Thus, the soft multi-pivot construction preserves invariance while reducing sensitivity to the singularity of any single pivot. The temperature $\tau$ controls the sharpness of pivot selection: small $\tau$ approximates hard selection of the most reliable pivot, while larger $\tau$ gives a smoother average over charts.

The weighted average should be viewed primarily as an implementation-level stabilization device. Since averaging different orbit representatives can, in principle, introduce collisions, one may append the weights to retain information about the selected chart:

$$\widetilde{z}_{\mathrm{soft}}(x) = \big(z_{\mathrm{soft}}(x), \alpha_1(x), \ldots, \alpha_m(x)\big).$$

Alternatively, one may use the concatenated representation $z_{\mathrm{cat}}$ when preserving maximal chart information is more important than compactness.

### H.3. Handling Odd-n and Non-Origin Rotations

A significant strength of our framework is its extensibility to cases where the data does not follow simple origin-centered symmetries or where the input dimension $n$ is odd.

For the odd-$n$ case, a rotation in $SO(n)$ necessarily leaves at least one dimension invariant. Let the input $x$ be projected into the coordinate system defined by $A$, such that $v = Ax = \bigoplus_{i=1}^{\lfloor n/2 \rfloor} v_i \oplus y_s$, where each $v_i \in \mathbb{R}^2$ and $y_s$ is the scalar invariant component $(a_s^T x)$. The invariant representation is defined by appending this invariant directly to the canonicalized blocks:

$$\text{invRep}(x) := \left( \|v_1\|_2 \mathbf{e}_1 \oplus \bigoplus_{i=2}^{\lfloor n/2 \rfloor} R(-t_0 \lambda_i) v_i \right) \oplus y_s \qquad (24)$$

This ensures that the information along the invariant axis is preserved while the rotational components are canonicalized onto the orbit representatives.

For non-origin rotations (assuming $n$ is even), let $c = \bigoplus_{i=1}^{n/2} c_i$ be the learned center of rotation. We define the centered coordinates as $v = A(x - c) = \bigoplus v_i$. The invariant representation must account for the offset to remain orbit-separating. The canonical orbit representative in the shifted coordinate space is:

$$\text{invRep}(x) := \left( \|v_1\|_2 \mathbf{e}_1 \oplus \bigoplus_{i=2}^{n/2} R(-t_0 \lambda_i) v_i \right) + Ac \qquad (25)$$

By learning the orientation $A$, rates $\lambda$, and center $c$ simultaneously, $H_\gamma$-Net discovers symmetries that are physically offset or embedded in odd-dimensional feature spaces.

### H.4. Richness of Geometric Orbits

A common misconception in symmetry discovery is that orbits under the action of one-parameter rotational subgroups are always simple circles. In reality, circular orbits are a special case that occurs only when rotation is confined to a single 2D plane (i.e., $\lambda_1 = 1, \lambda_2 = \cdots = \lambda_{n/2} = 0$). In the general elliptical case, the subgroup action involves simultaneous rotations across multiple 2D planes with potentially different integer rates $\lambda_i$. This produces rich, non-circular one-dimensional closed curves, analogous to high-dimensional Lissajous curves, that twist through $\mathbb{R}^n$.

Our framework's ability to handle these nuanced geometries is a direct result of the $\text{invRep}_G$ mapping. Simply taking the norm of each 2D block in the transformed space $v = Ax$ (i.e., $\|v_i\|$) or taking the norm in only one plane while leaving others unchanged does not produce a complete invariant representation for a 1D subgroup. The former case corresponds to invariance for an $n/2$-dimensional Lie subgroup and hence does not separate orbits in a 1D subgroup (for $n \geq 4$). Our modular $t_0$-based alignment ensures that we recover a representation that is both invariant and *orbit-separating*, a property we have theoretically guaranteed across elliptical, hyperbolic, and parabolic regimes.

### H.5. Avoiding the No-Symmetry Collapse

A central challenge in symmetry discovery, as highlighted in the Augerino framework (Benton et al., 2020), is the risk of a "no-symmetry" or trivial solution. In Augerino, the model learns a distribution $\mu_\theta$ over augmentations. Since standard training objectives select for flexibility rather than constraint, the optimizer may favor a distribution that collapses to a delta function at the identity, effectively recovering a standard non-symmetric model. Augerino mitigates this by adding an explicit regularization term to the loss to force a broader distribution.

In contrast, our framework inherently avoids the no-symmetry collapse through its architecture. By restricting the hypothesis space to functions of the form $f = \phi \circ \text{invRep}_G$, every representable function is provably invariant to some one-parameter subgroup $H_\gamma$. A non-symmetric function simply does not exist within the search space of the network. Consequently, even with infinite data, the model cannot collapse to a trivial solution but must converge to the most consistent symmetry present in the dataset.

### H.6. Overfitting and Lipschitz Analysis

Overfitting with an incorrect symmetry is architecturally penalized because it forces the mapping $\phi$ to learn an explosive complexity. Suppose the model selects a "wrong" symmetry that maps two distinct input points $x_1$ and $x_2$ (belonging to

distant ground-truth orbits) to nearby points in the invariant representation space, such that $\| \mathrm{invRep}(x_1) - \mathrm{invRep}(x_2) \| < \epsilon$. If these points have significantly different target values in the actual data manifold, such that the distance between ground-truth labels is $\| y_1 - y_2 \| > M$, the local Lipschitz constant $L$ of the network $\phi$ must satisfy:

$$L \geq \frac{\|\phi(\mathrm{invRep}(x_1)) - \phi(\mathrm{invRep}(x_2))\|}{\| \mathrm{invRep}(x_1) - \mathrm{invRep}(x_2)\|} > \frac{M}{\epsilon} \tag{26}$$

Forcing a network to learn such an explosive local mapping is computationally difficult for standard architectures and capacity constraints. This learning conflict results in a high training loss, which acts as a natural signal to steer the optimizer away from incorrect symmetries and toward the true $H_\gamma$. While validation loss is used as a reliable metric for detecting this divergence, practitioners can further stabilize training by adding regularizers to control the Lipschitz constant of $\phi$.

