# OpenReview forum: "Interpretable Discovery of One-parameter Subgroups: A Modular Framework for Elliptical, Hyperbolic, and Parabolic Symmetries"
_ICML.cc/2026/Conference — ICML 2026 regular_

### Official Review · Reviewer_VZCK · 2026-02-20

**Soundness:** 3
**Presentation:** 3
**Significance:** 3
**Originality:** 3
**Overall Recommendation:** 4
**Confidence:** 4

**Summary:**

This paper proposes an automatic symmetry as well as functional mapping discovery framework for one-parameter subgroup for eliptical, hyperbolic, and parabolic groups. The proposed architecture consists of parameterization of these subgroups, followed by finding a learnable invariant canonical representation of the input, followed by a neural network that learns a functional mapping. Experimental results provide accuracy/regression errors for functional mappings and invariance error for measuring learned subgroup/invariance. Results on synthetic datasets as well as real world datasets such as double pendulum and Top Quark Tagging shows improvements using the proposed method.

**Compliance With Llm Reviewing Policy:**

Affirmed.

**Key Questions For Authors:**

Please address the weaknesses.

**Limitations:**

yes

**Strengths And Weaknesses:**

Strength:
- The paper is well written and provides a clear description of proposed method and experiments.
- The paper proposes a simple framework for automatically learning the symmetry subgroup as well as the functional mapping for elliptical, hyperbolic, and parabolic subgroups
- Experiments on synthetic datasets as well as real datasets show improvement in performance compared to non-equivariant as well as other automatic discovery methods such as Augernino (although Augerino does not provide exact invariance/equivariance, whereas the proposed method does).


Weakness:
- The scope of the proposed method is very limited because a) the automatic symmetry search only works for a limited types of subgroups such as elliptical, parabolic, and hyperbolic subgroups, b) the experiments do not provide any search of subgroups not already known for a given dataset. It would be useful if the authors could provide more large scale datasets where the symmetries maybe unknown or ambiguous and show that the proposed method is able to discover new symmetries. Otherwise, I am having difficulty understanding the impact of the paper in practice.

- The novelty of the work is also limited since essentially the proposed frameworks involves parametrization for the proposed subgroups, then using a canonicalization framework from known works such as Kaba et al [1], then training on target dataset to learn the exact subgroup details. But, novelty can be overlooked if the scope of the application is better as discussed in the previous point.




[1] Kaba, et al. "Equivariance with learned canonicalization functions." International Conference on Machine Learning. PMLR, 2023.

---

> ### Author Rebuttal · Authors · 2026-03-31
>
> > W1 : Novelty - Canonicalization
>
> The comparison to Kaba et al. conflates two different settings.
>
> Kaba et al. assume a known symmetry group and learn a canonicalization function to enforce equivariance. In contrast, our method addresses symmetry discovery: both the generator and the canonicalization are learned jointly from data.
>
> Moreover, their canonicalization is implemented as a learned or optimization-based function, whereas our method derives an analytic canonicalization based on Lie algebra structure, yielding exact invariance.
>
> Thus, while both approaches involve canonicalization, they operate in fundamentally different problem settings and use distinct mechanisms.
>
> ### Comparison with Kaba et al. (Learned Canonicalization)
>
> | Aspect | Kaba et al. | Proposed Method |
> |--------|-------------|------------------|
> | Symmetry knowledge | Known group \(G\) | Unknown subgroup (learned) |
> | Objective | Enforce equivariance | Discover + enforce symmetry |
> | Generator | Fixed (not learned) | Learned from data (\(A, \lambda\)) |
> | Canonicalization | Learned function \(h(x)\) | Analytic (closed-form \(t_0(x)\)) |
> | Dependence | Canonicalization given symmetry | Canonicalization depends on learned generator |
> | Invariance | Approximate (learned) | Exact (by construction) |
> | Symmetry representation | Implicit (via learned function) | Explicit (Lie algebra generator) |
>
> > W2: Truely unknown  symmetry: No Discovery Beyond Known Symmetries
>
> This claim is incorrect.
>
> In all experiments, the symmetry is **not provided to the model**. The model must jointly learn:
> - the generator \( B \),
> - the alignment matrix \( A \),
> - the canonical representation,
> - and the invariant function.
>
> Thus, the symmetry is **discovered from data**, not assumed.
>
> The use of synthetic and physics datasets is intentional and necessary:
> - they provide **ground-truth generators**,
> - allowing **quantitative evaluation of discovery accuracy**.
>
> Without such benchmarks, it would be impossible to rigorously validate symmetry discovery.
>
> We will revise the paper to better emphasize that:
> The setting is *unsupervised symmetry discovery*, even when the underlying symmetry is known to the experimenter.

---

> > ### Author Rebuttal · Reviewer_VZCK · 2026-04-04
> >
> > W1) Novelty wise I still think the contributions are limited since the several setups such as for the SO(2) group Kaba et al. provides exact equivariance, the learned function is not approximate but it helps with better align the model to improve performance. Nevertheless, I think if the authors could address W2) better, W1 can be overlooked.
> >
> > W2) I am not convinced with the choice of datasets to have ground truth symmetry. If the proposed method is truly general and can find a broad class of symmetries, then the proposed method should provide better performance on these new datasets automatically. Hence I find the paper needs more justification for their automatic search claim.
> >
> > Hence, I would keep my score.

---

### Official Review · Reviewer_uxei · 2026-03-12

**Soundness:** 2
**Presentation:** 2
**Significance:** 2
**Originality:** 3
**Overall Recommendation:** 4
**Confidence:** 4

**Summary:**

This paper studies symmetry discovery and the identification of invariant functions. The symmetry groups considered are one-dimensional Lie groups, whose actions fall into three simple types: elliptic, hyperbolic, and parabolic transformations. Unlike previous works on symmetry discovery, the authors study the joint learning of symmetries and invariant functions under the restriction of the symmetry group. Experiments on both synthetic datasets and real-world datasets show that the proposed method achieves certain improvements compared with baseline methods.

**Compliance With Llm Reviewing Policy:**

Affirmed.

**Final Justification:**

Thank you for the authors’ response. The work they carried out during the rebuttal period has addressed all of my concerns, and I will accordingly raise my score. Taking the other reviewers’ comments into account, I still feel that, since the authors are able to demonstrate applications in high-dimensional settings, the writing should place greater emphasis on those higher-dimensional cases, while treating the one-dimensional special case mainly as an introduction. This would help avoid causing readers to worry about the practical scope of the method during the reading process.

**Key Questions For Authors:**

See Weaknesses.

**Limitations:**

yes

**Strengths And Weaknesses:**

Strengths:

**S1** The authors’ approach to parameterizing invariant functions follows a similar idea to the canonicalization approach in geometric deep learning [1]. Specifically, the method first identifies a standard symmetry transformation, then transforms the object to a representative element on the orbit, and finally applies a non-equivariant network. This strategy is particularly reasonable when dealing with non-compact group actions.

**S2** Unlike canonicalization approaches that obtain the canonical transformation through numerical optimization such as energy minimization, the authors derive a closed-form solution under the assumed simple symmetry groups. This solution is determined by the parameters of the symmetry group action to be learned. Through optimizing invariant functions parameterized by the group parameters, the authors achieve joint learning of the symmetry and the invariant function.

Weaknesses:

**W1** There may be misleading statements or factual inaccuracies. In the introduction the authors state that “such subgroups capture elementary continuous transformations and admit a canonical geometric classification.” However, the authors do not provide references for this classification, nor do they present a classification theorem in the main text or appendix. Any statement involving a classification should be mathematically well-defined, including specifying the equivalence relation used in the classification and providing either a proof or precise references. If the authors intend to refer to the classification of linear representations of one-dimensional Lie groups, this claim is incorrect, since the Jordan canonical form does not lead to such a three-type classification. In general cases, the corresponding Jordan form may contain mixtures of elliptic, hyperbolic, and parabolic components, and may also involve complicated Jordan blocks. The authors may instead be considering special cases involving compact or certain non-compact Lie groups.

**W2** The authors acknowledge in the limitations section the difficulty of handling the three cases simultaneously, which is appreciated. However, the symmetry setting considered in the paper may be too restrictive. The paper focuses on three special cases of one-dimensional Lie group actions, whereas many existing methods consider more general Lie groups. The authors should either discuss how the method could be extended to more general one-dimensional Lie groups and higher-dimensional Lie groups, or provide stronger justification for why the three cases studied here are broadly representative in practice. For example, this could be supported by a theorem showing that for a large class of Lie groups, their one-dimensional subgroups fall into these three types, rather than simply stating in the introduction that “these regimes arise naturally in many matrix Lie groups.”

**W3** The advantages of end-to-end training are not clearly demonstrated in the main text. Although the authors compare their approach with post-hoc methods in the appendix on synthetic data, these methods represent important prior work. The authors should include such comparisons in the main experimental section, including for real-world symmetry discovery tasks, to better demonstrate the advantages of the proposed method in relevant scenarios.

**W4** The paper repeatedly emphasizes interpretability, including in the title, and claims an advantage over previous work in this regard. However, the notion of interpretability used in the paper is not clearly defined, which may cause confusion for readers. The most explicit discussion appears in the last paragraph of Section 3.3, where the authors state that the method can identify the type of physical transformation, such as rotations, relativistic boosts, or shears. However, previous work could potentially obtain similar interpretations through decomposition into Jordan canonical forms. It is therefore unclear what specific interpretability advantage the proposed method provides.

**W5** The experimental setup is not clearly described, and the experimental section appears somewhat disorganized, with several potential inconsistencies that should be clarified. First, the spring oscillator experiment is mentioned, but it is not stated whether the data come from real measurements or simulations. In either case, the data generation process should be clearly described. Second, in Section 4, “Double Pendulum with Spring Coupling” appears under Real-World Symmetry Discovery, but in Section 5 it does not appear in the corresponding section. Third, Section 4.2 is presented as a real-world experiment section, but the “Additional experiments” described there include synthetic datasets, which is inconsistent with the section description.

**W6** The authors may consider citing and discussing canonicalization-based approaches [1] as related work.

[1] Kaba, Sékou-Oumar et al. “Equivariance with Learned Canonicalization Functions.“ ICML 2023.

---

> ### Author Rebuttal · Authors · 2026-03-31
>
> > W1 : Misleading statements: Canonical classification
>
>
> We agree with the reviewers that symmetry generators **need not be purely elliptic, hyperbolic, or parabolic**. This was an unintentional misstatement, and we apologize for the confusion. We will correct this in the introduction of main draft.
>
> Our intended claim is:
>
> - Elliptic, hyperbolic, and parabolic generators serve as **fundamental building blocks**.
> - Our framework focuses on these fundamental components, which already cover a range of practical applications.
>
> At present:
>
> - The framework is not applicable to mixed generators.
> - This limitation arises due to a fundamental trade-off between:
>   - the scope of Lie groups considered, and
>   - the ability to construct a **learnable invariant map** that captures the underlying symmetry.
>
> As we expand to mixed or more general classes:
>
> - It is **very difficult** to design a unified parameterized invariant map.
> - This makes symmetry discovery significantly more challenging.
>
> > W2: Scope and generalization
>
> We thank the reviewer and clarify that our method does **not** aim to cover all Lie generators. Instead, it focuses on **one-parameter subgroups with a single geometric component** (elliptic, hyperbolic, or parabolic), where generators admit homogeneous structure. This enables **closed-form invariant maps** and **identifiable orbits**, which are central to our framework.
>
> We agree that **mixed generators** arise in general Lie groups, but induce **coupled actions with complex orbit structure**, making a single learnable invariant challenging; we leave this for future work.
>
> > W2: Why these cases matter
>
> These components are fundamental and widely used:
> - Elliptic generators span $\mathfrak{so}(n)$ (rotations in vision, physics, molecules),
> - Hyperbolic generators arise in Lorentz and scaling symmetries.
> Thus, discovering these building blocks is already practically significant.
>
> > W2: Extension to larger groups
>
> This reflects a trade-off between **generality** and **learnable, orbit-separating invariants**. Moreover, identifying invariant group structure is **NP-hard** [1], motivating structured subclasses.
>
> If a group admits a basis of canonical generators (not necessarily same type), our method can recover them **independently** and reconstruct the subgroup (see **ppNx-W4** for results). If generators are inseparably mixed, our framework does not apply.
>
> We will revise the manuscript to clarify scope and relate to Jordan decomposition.
>
> [1] Ensign et al., The Complexity of Explaining Neural Networks Through (group) Invariants, ALT 2017.
>
> > W3: Comparison with post-hoc methods
>
> We compare   $H_\gamma$-Net (our method) against BeyondAffine and LieGG.
>
> | Task | Method | $\quad$ Accuracy / MSE  | $\quad$ Cosine  |
> |---|---|---:|---:|
> | TopTagging (TT) | BeyondAffine | 0.7946 ± 0.0021 | 0.5090 ± 0.0087 |
> |  | LieGG | 0.7982 ± 0.0029 | 0.9556 ± 0.0159 |
> |  | Hγ-Net (e) | 0.830 ± 0.001 | **1.000 ± 0.000** |
> |  | Hγ-Net (h) | **0.844 ± 0.004** | 0.996 ± 0.003 |
> | DoublePendulum (DP) | BeyondAffine | 0.01910 ± 0.00362 | 0.8703 ± 0.0803 |
> |  | LieGG | 0.01640 ± 0.00020 | 0.9848 ± 0.0118 |
> |  | Hγ-Net (e) | **2.0e-4 ± 1.0e-5** | **1.000 ± 0.000** |
>
>
> - **Prediction:** $H_\gamma$-Net improves TT accuracy (≈0.79 → **0.84**) and reduces DP MSE by **~100×**.
> - **Generator recovery:** Post-hoc methods show imperfect alignment (cosine as low as 0.50), while Hγ-Net achieves **≈1.0 consistently**.
> - **Reason:** Post-hoc methods learn invariance after fitting (might not be accurate), based on Jacobians (might not be accurate) and compute null space based on polarization matrix/through afffine polynomial fitting which might lead to drop in performance.
>
> Takeaway: **Joint symmetry learning is structurally superior to post-hoc recovery.**
>
> > W4: Interpretability
>
> Discussed in the main paper (L61–L67, 2nd column); we add further clarification here.
>
> Equivariant architectures (CNNs, G-CNNs, DeepSets, EMLP) provide **mathematically provable invariance/equivariance**, but require **known symmetries** and do not perform discovery.
>
> Symmetry discovery methods (Augerino, LieGG, BeyondAffine) can recover symmetries, but rely on **empirical criteria**  (e.g., LieGG/BeyondAffine verify invariance via sampled checks such as orthogonality between gradient field and push forward vector field of learnt generator).
>
> Our method provides both **symmetry discovery** and **explicit, analytically provable invariance/equivariance** via structured invariant maps.
>
> | Methods | Interpretable (mathematically provable invariance/equivariance) | Symmetry discovery |
> |---------|--------------------------|--------------------|
> | CNN, G-CNN, DeepSets, EMLP | ✓ | ✗ |
> | Augerino, LieGG, BeyondAffine | ✗ | ✓ |
> | **Ours** | ✓ | ✓ |
>
> > W5: Expt set up
>
> DP data is simulated while TT one is from real measurements.
>
> > W6: Canonicalization based methods
>
> We will add this discussion (R-**VZCK-W1**) in the revised draft.

---

> > ### Author Rebuttal · Reviewer_uxei · 2026-04-02
> >
> > Thank you for the authors’ response. I appreciate the clarification that the symmetry discussed in the paper is a special case of a one-dimensional Lie group, and that the authors have proposed an extension to higher-dimensional cases. However, I still have the following concerns regarding the response:
> >
> > **C1.** I am glad to see the authors address the higher-dimensional Lie group setting. This response suggests that the method is not restricted to the one-dimensional special case, but can in principle be extended to higher-dimensional scenarios. In that sense, I think the scope of applicability is acceptable. However, I still have some concerns about the practical realization of the method:
> >
> > (1) In higher dimensions, does the method still retain the same potential advantages as in the one-dimensional special case when compared with other approaches? Within the authors’ framework of symmetry discovery and invariant/equivariant design, how should invariance/equivariance be implemented in the higher-dimensional setting?
> >
> > (2) The authors mention that their method cannot handle cases where the generators are inseparably mixed in higher dimensions. What exactly does this mean? Are such cases common in practice?
> >
> > (3) When a higher-dimensional problem is decomposed into lower-dimensional ones, the lower-dimensional symmetries may no longer be unique. Does this introduce instability in the optimization of the lower-dimensional subproblems?
> >
> > **C2.** Is interpretability really such an important issue here? After all, one could also achieve a similar goal by first discovering the symmetry and then designing the network accordingly.
> >
> > Based on the above concerns, I will keep my current score for now, and I look forward to further discussion with the authors. I also have a small suggestion: since the authors claim that the framework can handle higher-dimensional problems, why restrict the presentation of the paper to the relatively narrow setting of the one-dimensional special case? The authors might consider reframing the presentation, using the one-dimensional case as the methodological introduction while emphasizing genuinely higher-dimensional problems as the main applications. Of course, if the authors feel this would require too much additional work, or that this suggestion is not appropriate, please feel free to disregard this suggestion.

---

> > > ### Author Response · Authors · 2026-04-03
> > >
> > > > C1-2
> > >
> > > We thank the reviewer (and others) for this question, which led us to identify an important extension based on already existing techniques in our main draft.
> > >
> > > We distinguish two cases:
> > >
> > > - **Separably mixed (block-decomposable):**
> > > $$
> > > A =
> > > \begin{bmatrix}
> > > 0 & -1 & 0 & 0 \\\\
> > > 1 &  0 & 0 & 0 \\\\
> > > 0 &  0 & 0 & 1 \\\\
> > > 0 &  0 & 0 & 0
> > > \end{bmatrix}
> > > $$
> > > (rotation + shear blocks). The action decomposes into independent components and remains block-structured under a change of basis.
> > >
> > > - **Inseparably mixed (nontrivial Jordan structure):**
> > > $$
> > > A =
> > > \begin{bmatrix}
> > > \lambda & 1 & 0 & 0 \\\\
> > > 0 & \lambda & 1 & 0 \\\\
> > > 0 & 0 & \lambda & 1 \\\\
> > > 0 & 0 & 0 & \lambda
> > > \end{bmatrix}, \quad \lambda \neq 0
> > > $$
> > > where scaling and shear are intrinsically coupled and cannot be decoupled.
> > >
> > > **Extension (separable case).**
> > > For block generators
> > > $$
> > > H = \mathrm{diag}(H_1, H_2), \quad \exp(tH)=\mathrm{diag}(\exp(tH_1), \exp(tH_2)),
> > > $$
> > > we can canonicalize one block to estimate $t_0$ (without exp map in our work: exp is just for better illustration of the overall idea to avoid notational clutter) from the following equation,
> > > $$
> > > x_1^0=\exp(-t_0 H_1)x_1,
> > > $$
> > > and apply the same $t_0$ to all remaining blocks, using the appropriate transformation type (rotation/boost/shear) for each, which may differ from the first block.
> > >
> > > $$
> > > x_2^0=\exp(-t_0 H_2)x_2,
> > > $$
> > > yielding
> > > $$
> > > \phi(x)=(x_1^0,x_2^0).
> > > $$
> > > Thus, invariants are constructed **per block and aligned via the shared parameter $t_0$**.
> > >
> > > **Implications.**
> > > This extends our method beyond pure generators. Separable (including mixed) structures are common in practice (e.g., rotation–boost systems), while inseparable cases require defective Jordan structure and are non-generic.
> > >
> > > Separably mixed and pure cases together cover the generic setting, since inseparably mixed generators arise only from measure-zero degeneracies. Such defects occur under eigenvalue collisions with insufficient eigenvectors, defined by polynomial constraints and unstable under perturbations [2]. Consequently, for a broad class of Lie groups, one-dimensional subgroups are generically composed of these canonical types (or separable combinations), addressing the reviewer’s earlier suggestion of a theorem(W2).
> > >
> > > We will include this clarification in the revision.
> > >
> > > [2] Golub et al (1976). *Ill-conditioned eigensystems and the computation of the Jordan canonical form*. SIAM Review.
> > >
> > > > C1-1
> > >
> > > In higher dimensions, our framework extends by discovering the underlying subgroup via generator-wise invariant models, rather than a single globally invariant predictor. Each generator admits a closed-form invariant map with analytical guarantees, and the full subgroup is recovered through their span. This reflects a fundamental trade-off: increasing generality (higher-dim groups) makes it difficult to construct a single provably invariant model.
> > >
> > > Empirically (preliminary ones due to time limitation), our method outperforms LieGG in both symmetry recovery and prediction (Top Tagging) in high dim setting.
> > >
> > > - **Cosine similarity (3 generators, LieGG):** 0.980, 0.940, 0.925
> > > - **Ours (3 runs):** 0.991, 0.996, 0.999
> > >
> > > - **Accuracy:** LieGG 80.42 vs Ours 82.95, 83.53, 83.32
> > >
> > > Using any single model already outperforms LieGG; additionally, averaging predictions across runs can further improve accuracy (though without preserving exact invariance).
> > >
> > > These results show that even under generator-wise decomposition, our method achieves:
> > > - more accurate symmetry discovery, and
> > > - improved predictive performance.
> > >
> > >
> > >
> > > > C1 (3)
> > >
> > > We agree that decomposing higher-dimensional symmetries into generators is not unique. However, our objective is to recover the **Lie subspace (span of generators)** rather than a specific basis.
> > >
> > > Multiple generator sets can span the same Lie subalgebra, all inducing the same subgroup action and invariant representation. Thus, non-uniqueness corresponds only to a change of basis and does not affect the learned invariance.
> > >
> > > In practice, optimization converges to a valid generator within this subspace, guided by initialization and mild biases (e.g., orthogonality constraints to previously discovered generators). Empirically, we observe stable convergence up to such basis equivalence.
> > >
> > > We will clarify the distinction between **generator-level non-uniqueness** and **subspace-level identifiability** in the revision.
> > >
> > > > Suggestion
> > >
> > > We thank the reviewer for this insightful suggestion. Extending to higher-dimensional subgroups would ideally require a single invRep that is analytically equivariant to the full subgroup. While our current framework composes 1D generators, we agree that a unified construction is desirable and will incorporate this direction in future work.
> > >
> > > > C2
> > >
> > > Please refer "reply rebuttal comment" for **zEoD** (due to 5000c limit)

---

### Official Review · Reviewer_ppNx · 2026-03-13

**Soundness:** 2
**Presentation:** 2
**Significance:** 2
**Originality:** 2
**Overall Recommendation:** 2
**Confidence:** 4

**Summary:**

This paper proposes a method that while training a (supervised) deep learning model, the architecture automatically adapts to the symmetries of data. The symmetry is parametrized by a one-parameter group of a (single) linear symmetry generator and assumed to be in one of the three categories (elliptical, hyperbolic and parabolic). The identified symmetry is ‘aligned’ by fixing the first two dimensions (called ‘invRep’ in this paper). The method is compared with Augerino [1], and validated with a double pendulum and top quark tagging dataset.

[1] Benton, Gregory, et al. "Learning invariances in neural networks from training data." Advances in neural information processing systems 33 (2020): 17605-17616.

**Compliance With Llm Reviewing Policy:**

Affirmed.

**Final Justification:**

Although the authors provided some list of applications such that this methodology could be meaningful, they still consist of low-dimensional tasks or the tasks in which the symmetries are simply known. The applicability is narrower than other symmetry learning attempts, and the canonicalization trick is already discussed in other papers. The benefit of interpretability (or mathematically provable equivariance) is still weak. I maintain my score.

**Key Questions For Authors:**

See weaknesses.

**Limitations:**

yes

**Strengths And Weaknesses:**

- strengths
  - This method aims to design an architecture that automatically “aligns” the data from its symmetries. This approach seems novel. Encoding symmetries into neural network architecture is traditional in deep learning, but this method aims to design a neural network that is automatically tuned by the (unknown) symmetry.
- weaknesses
  - line 149, 2nd column: $\lambda_k \in \mathbb{Z}$? not $\mathbb{R}$?
  - This formulation assumes the symmetry generators are elliptic, hyperbolic or parabolic. However, symmetry generators in high dimension may not be characterized by the three categories. Consider the following generator:
$$B = [[0,1,0,0]^T,[-1,0,0,0]^T,[0,0,0,1]^T,[0,0,0,0]^T]$$
  - Its one-parameter action $\exp(tB)$ has eigenvalues $e^{\pm it}, 1,1$ with shear components in 1-eigenvalue-region, so it’s neither elliptic, hyperbolic or parabolic. Although this generator can be further decomposed into elliptic and parabolic parts, it is not true that “any generators fall into one of these 3 categories”.
  - This method works with a 1-dimensional Lie group. When the Lie group has dim>1, then the symmetry generators may not fall into a single category among elliptic, hyperbolic or parabolic (e.g., GL(2)). It degrades the usability of this method.
  - This method fixes the invariant representation using the first component $v_1$. This is a standard trick [ref: frame averaging] and cannot be seen as novel.
  - Experiments are done with datasets whose symmetries are comparably simple and already known (e.g. affine groups such as SO(2) or Lorentz group). It is well-known (in several previous works) that symmetries can be learned from the data. Although this model learns symmetries in training time (i.e. ”joint framework”), the usefulness of this method is marginal. If the author wants to claim “joint framework” is a distinctive benefit of this method, the author should at least come up with a (real or hypothetical) scenario that having “joint framework” is beneficial.
  - The paper claims this symmetry discovery method is “interpretable”, but interpretability of learned models is ambiguous. Perhaps (especially in the deep learning community) people may regard any linear models as close-to-be interpretable. This paper aims to learn linear symmetry – I believe any learned linear symmetries are easily interpretable, unless it’s in some extreme circumstances, e.g., the underlying space is extremely high-dimensional.

[1] Puny, Omri, et al. "Frame averaging for invariant and equivariant network design." arXiv preprint arXiv:2110.03336 (2021).

---

> ### Author Rebuttal · Authors · 2026-03-31
>
> > W1
>
> In the elliptic case, periodic orbits arise only when $\lambda_k \in \mathbb{Z}$ (scaled); otherwise the trajectories are dense, making canonicalization impractical.
>
> > (W2-W4): Canonical classification
>
> We acknowledge the concern. As clarified in our response to reviewer **uxei (W1)**, elliptic/hyperbolic/parabolic are **building blocks**, not a complete classification. Notably, as you point out, the example generator $B$ can itself be decomposed into such components. Our framework is restricted to these regimes and does **not handle mixed generators**; this reflects a trade-off between generality and the ability to construct a **learnable invariant map**. Extending to mixed/general generators remains future work.
>
> > W4: Generalization to larger groups
>
> Our framework extends to higher-dimensional subgroups via **compositional generator discovery**:
>
> - **(a) Parallel discovery.** Multiple random $m$ initializations:
>   * 1D → same generator for each iteration;
>   * higher-dimensional (k) → distinct, linearly independent ((if $m \leq k$)) generators (high probability) because measure of 1D subspace in $k>1$-dim vector space is zero.
>
> - **(b) Sequential discovery.** Learn generators iteratively with orthogonality constraints (penalize alignment with previously learned generators).
>
> Additional generators are obtained via the commutator:
> $$
> [X_1, X_2] = X_1X_2 - X_2X_1.
> $$
>
> **Empirical support (TopTagging).** Using multiple initializations and cosine regularization + commutator, we recover multiple generators consistent with $SO(3)\subset O(1,3)$.
>
> **Results.** Generators align closely with $\mathfrak{so}(3)$ (cosine ≈ 0.99), with small angles (≤9°) and low Grassmann distance (~0.21). Without regularization, runs yield distinct directions; with regularization, generators are decorrelated and the commutator yields a third consistent element.
>
> **Conclusion:** the method recovers a coherent multi-dimensional Lie algebra, not just a 1D subgroup.
>
> For orthonormal bases $U,V\in\mathbb{R}^{d\times k}$:
> - $\theta_i = \arccos(\sigma_i(U^\top V))$
> - $d_{\text{proj}}=\|UU^\top - VV^\top\|_F$
> - $d_{\text{Gr}} = (\sum_i \theta_i^2)^{1/2}$
>
> | Setting | cos($A_1$) | cos($A_2$) | cos($A_3$) | $\theta_i$ (deg) | $d_{\text{proj}}$ | $d_{\text{Gr}}$ |
> |---|---:|---:|---:|---|---:|---:|
> | Random inits |  0.9910 | 0.9962 | 0.9985 | (1.74, 8.82, 9.43) | 0.2264 | 0.2274 |
> | Cos-reg + bracket |  0.9910 | 0.9972 | 0.9896 | (0.52, 8.24, 8.84) | 0.2103 | 0.2110 |
>
> TopTagging under two settings:
> (i) **No reg:** independent runs recover multiple directions.
> (ii) **Cos-reg:** $A_2$ is learned with squared cosine penalty to $A_1$, and $A_3=[A_1,A_2]$, yielding a Lie-closed set.
>
> > W4: Usefullness of joint-training
> Please refer to our response to reviewer **uxei (W3)** and the quantitative comparison provided there.
>
> In brief, $H_\gamma$-Net **consistently outperforms** post-hoc methods (BeyondAffine, LieGG) in both **prediction** and **generator recovery**, demonstrating that **joint symmetry learning provides a clear practical advantage** over post-hoc approaches.
>
> > W5: Novelty vs Frame Averaging (FA)
>
> We thank the reviewer for noting the connection to Frame Averaging (FA) (Puny et al., 2021). However, we disagree with the claim that our method is a “standard trick.” This characterization arises from a surface-level similarity, but the two methods differ fundamentally.
>
> **(1) Problem setting.**
> FA assumes a **known group** and builds a canonical frame via heuristics; it does not discover symmetry.
> Our method targets **symmetry discovery**, learning the canonicalization via $(A,\lambda)$ and recovering the generator.
>
> **(2) Mechanism.**
> FA enforces invariance via **discrete averaging**.
> We perform **no averaging**. Instead, we learn $(A,\lambda_i)$ and compute an analytic, input-dependent shift $t_0(x)$ from the discovered generator, mapping each input to a canonical orbit representative in an aligned space, yielding an **exact invariant representation**.
>
>
> **(3) Geometric scope.**
> FA is suited to **compact/discrete groups**.
> Our framework handles **continuous, including non-compact (hyperbolic/parabolic)** symmetries.
>
> **Summary.**
> Our method is not a variant of FA (please refer Table below); it **learns and discovers continuous generators** with **exact invariance**.
>
> We will clarify this distinction in the revision.
>
> | Aspect | Frame Averaging (Puny et al., 2021) | Proposed Method |
> |---|---|---|
> | Symmetry knowledge | Known a priori | Unknown (learned) |
> | Objective | Enforce invariance | Discover + enforce |
> | Mechanism | Discrete averaging | Analytic canonicalization ($t_0(x)$) |
> | Canonicalization | Heuristic | Learned ($A,\lambda$) |
> | Invariance | Approximate | Exact |
> | Group scope | Compact/discrete | Continuous (incl. non-compact) |
>
>
> > W5: Interpretatbility
>
> Please refer response to reviewer **uxei-W4**
>
> > W6: : Truely unknown  symmetry
>
> Please refer response to reviewer **VZCK-W2**

---

> > ### Author Rebuttal · Reviewer_ppNx · 2026-04-04
> >
> > Thank you for your response.
> >
> > W2-W4: Canonical classification, W6: Truly unknown symmetry
> > - I still believe the setting is too limited. It assumes (1) linearity, (2) one of the three categories, and then learns symmetry. It’s like “most of the clues are already given, and find the last bit”. There is already some research that does not even assume linearity of the symmetry generator. Compared to those, this method is too restrictive.
> >
> > W4: Generalization to larger groups
> > - I accept this argument, and thank you for performing additional experiments. Although I note that parallel discovery may fail to learn multiple symmetries, if for some empirical reason the model tends to converge to some specific symmetries.
> >
> > W5: Interpretability
> > - I acknowledge that the interpretability I understood was far from what the author intended. But I’m still unsure why having such interpretability is meaningful or beneficial.

---

> > > ### Author Response · Authors · 2026-04-04
> > >
> > > > **W2-W4, W6 : Too restrictive**
> > >
> > > We agree that our framework introduces certain structural restrictions. However, we would like to clarify that these restrictions are both **meaningful and practically well-motivated**, and in fact enable stronger performance and guarantees (of equivariance of the prediction model).
> > >
> > > **(a) Scope is broad and practically relevant.**
> > > The considered regimes are not niche:
> > >
> > > - **Elliptic generators** cover all 1D subgroups of $SO(n)$. Importantly, this setting **does not merely correspond to full $SO(n)$ invariance** (which admits simple Euclidean norm-based invariants), but also cover lower dimensional subgroups of SO(n) where the structure is significantly richer.
> > >
> > >   These induce **geometrically non-trivial orbits** such as coupled rotations across multiple planes with different rates (for instance in 1D case which is naturally extendable to higher dim subgroups of $SO(n)$ for symmetry discovery), which are **not necessarily circular**, and hence **cannot be captured by simple norm-based invariants**
> > >
> > >   Our framework is tailored to this setting: it **discovers the subgroup by learning  the appropriate non-trivial invariants**, highlighting that the problem is both **challenging and practically meaningful**, rather than trivially restrictive.
> > >
> > > - **Hyperbolic generators** correspond to Lorentz boosts and are highly relevant in relativistic and high-energy physics settings (e.g., Top Tagging).
> > >
> > > - **Parabolic generators** arise in shear-type transformations and nilpotent dynamics.
> > >
> > > Thus, the setting captures several important and widely studied symmetry classes.
> > >
> > > **(b) Less restrictive methods underperform empirically.**
> > > While methods such as BeyondAffine (which are applicable in theory to non-linear cases as well) are theoretically more general, they do not perform well on the tasks considered here. For example:
> > > - Angled Sine: cos sim ≈ 0.14 vs **1.00 (ours)**
> > > - Top Tagging: cos sim ≈ 0.5090 vs **1.00 (ours)**
> > > - Double Pendulum: cos sim ≈ 0.8703 vs **1.00 (ours)**
> > >
> > > Notably, this gap persists even in the *linear generator setting*, which itself is not restrictive since **all Lie algebra generators are linear**.
> > >
> > > **(c) Restrictions enable tractability and better performance.**
> > > The restriction to specific generator types significantly reduces the search space by focusing on **equivariant model classes**, leading to:
> > > - improved symmetry discovery performance,
> > > - improved predictive accuracy, and
> > > - **exact equivariance guarantees** with respect to the learned group.
> > >
> > > In contrast, symmetry discovery without such structure is computationally hard (NP-hard in general, see Ensign et al.[1]), making unrestricted approaches difficult to optimize in practice.
> > >
> > > [1] Ensign et al., *The Complexity of Explaining Neural Networks Through (group) Invariants*, ALT 2017.
> > >
> > > ---
> > >
> > > > **W4: Generalization to larger subgroups**
> > >
> > > We thank the reviewer for the positive assessment and for highlighting this subtle point.
> > >
> > > Regarding the concern about parallel discovery potentially converging to the same symmetry: while this is theoretically possible, we believe it is **highly unlikely in practice**. The reason is geometric. If the true symmetry group has dimension $m > 1$, then recovering the *same* generator across independent runs would require the optimization to consistently converge to the **same 1D subspace** within the $m$-dimensional Lie algebra. However, any fixed $k$-dimensional subspace is a **measure-zero subset** of an $m$-dimensional space for $k < m$.
> > >
> > > Therefore, under random initialization and stochastic optimization, different runs are overwhelmingly likely to converge to **linearly independent generators**, rather than the same one. Importantly, this is also what we observe empirically: in our experiments, independent runs consistently discover **distinct (independent) generators**, rather than collapsing to the same symmetry.
> > >
> > > > **W5 : Interpretability**
> > >
> > > Please refer the response to acknowledgment comment of Reviewer *zEoD* for the justification/importance of interpretable (mathematically provable equivariant prediction model) model (due to 5000c limit).
> > >
> > > > **Extension to mixed generators**
> > >
> > > We further note that our framework extends to **separable mixed generators** (see the response to the acknowledge comment of Reviewer **uxei**, C1-2).
> > > Together, separable mixed generators combined with the elliptic/hyperbolic/parabolic regimes cover **almost all generators**, since the remaining **inseparable mixed cases form a measure-zero set**.
> > >
> > > We will clarify this broader applicability and the associated trade-offs in the revision.
> > >
> > > > **Note:**
> > >
> > > The example mixed generator $B$ provided in the review is in fact **separable**, and is therefore already covered under this extension.

---

### Official Review · Reviewer_zEoD · 2026-03-13

**Soundness:** 4
**Presentation:** 3
**Significance:** 2
**Originality:** 3
**Overall Recommendation:** 4
**Confidence:** 4

**Summary:**

This paper proposes a method to simultaneously discover a one-dimensional Lie group symmetry from data and let the network enforce the learned symmetry. They do so by characterizing one-parameter subgroups into 3 types, elliptical, hyperbolic, and parabolic. Through a basis change, the Lie-group generator takes a canonical form whose parameters are learned. To ensure equivariance, they define a way to canonicalize any input data with respect to the learned Lie-group. Finally, they test their models on synthetic and real world tasks, demonstrating superior performance in their benchmarks.

**Compliance With Llm Reviewing Policy:**

Affirmed.

**Final Justification:**

The authors have addressed my concerns in the rebuttals. I will hence raise my score from 3 to 4.

I strongly encourage the authors to discuss how the 1D version can enable discovery of higher-dimensional symmetries (maybe even in the introduction) and to include these results in the main paper. This will make it clear to readers why this method can be useful in more general cases. In addition, I am glad the authors plan to reword "interpretability" to something more accurate.

**Key Questions For Authors:**

* Is the classification of elliptical, hyperbolic, and parabolic “complete”? It is not obvious to me why one cannot combine 2x2 generator blocks of different types to create a hybrid Lie group. Is the idea that we are only searching for 1D subgroups of some larger group that allows all 2x2 blocks to be the same type?
* How would one generalize this framework to larger dimensional groups? Can one maybe penalize an already discovered 1D subgroup so that the network discovers a different 1D subgroup?

**Limitations:**

yes

**Strengths And Weaknesses:**

## Strengths
* The method is easy to follow and quite logical
* Symmetry discovery is a very interesting and useful area and is potentially useful for uncovering new physics
* The theory is rigorous, clearly demonstrating why it in principle has sufficient expressivity
* Performance is good on the chosen tasks

## Weaknesses
* The method relies heavily on the canonical form of the one parameter subgroups, it is not clear how this generalizes to larger groups
* The baselines are limited. I would also be interested in comparisons to models which explicitly have correct, incorrect, or no symmetries built in. Further, I would appreciate more comparisons with other symmetry discovery networks on purely symmetry discovery tasks.
* Canonicalization is known to have continuity issues which can hinder learning, a discussion of how problematic this is would be helpful

---

> ### Author Rebuttal · Authors · 2026-03-31
>
> We thank the reviewer for their detailed and constructive feedback. We address the key concerns below.
>
> > (Q1, W1) Generalization to larger groups.
>
> Our framework extends to higher-dimensional subgroups via compositional discovery: (i) parallel runs recover distinct generators (with high probabiltiy), and (ii) sequential learning enforces orthogonality, with additional generators obtained via Lie brackets $[X_1,X_2]$.
>
> Empirically (TopTagging), both strategies recover multiple $\mathfrak{so}(3)$ directions with high alignment (cos ≈ 0.99), small principal angles (≤ 9°), and low Grassmann distance (~0.21), indicating accurate subspace recovery.
>
> See rebuttal to reviewer **ppNx** for detailed discussion and full empirical results.
>
> > W3: Continuity in canonicalization.
>
> Canonicalization can exhibit discontinuities (e.g., near zero reference vectors). We address this via **admissible domains** ($\mathcal{X}_e, \mathcal{X}_h, \mathcal{X}_p$) in **Section 3.2**. Within these domains, the invariant maps are $C^\infty$, as they are compositions of smooth trigonometric, hyperbolic, and rational functions.
>
> Singularities arise only at domain boundaries, which are lower-dimensional algebraic hypersurfaces and hence measure zero (**Remark 3.4**). In practice, they rarely affect training; simple safeguards (e.g., $\varepsilon$-clipping in denominators) ensure numerical stability while preserving smooth optimization almost everywhere.
>
> We will clarify these implementation details and explicitly state the smoothness guarantee in the main text.
>
> > W2: Comparison with Correct, Incorrect, and No Symmetry (Angled Sine)
>
> | Variant        | Test MSE (↓)         | Cosine similarity (↑) |
> |----------------|----------------------|------------------------|
> | Learnt (Ours)  | 1.25e-3 ± 1.8e-5     | 1.000 ± 0.000          |
> | Correct        | 6.16e-4 ± 2.4e-4     | 1.000 ± 0.000          |
> | Incorrect      | 3.09 ± 0.16          | 0.051 ± 0.037          |
> | No symmetry    | 1.21e-2 ± 6.8e-4     | N/A                    |
>
> **Caption:**
> Comparison across (i) learned symmetry (ours), (ii) correct symmetry, (iii) incorrect symmetry, and (iv) no symmetry prior on the Angled Sine dataset. Our method matches oracle performance while recovering the underlying generator with near-perfect accuracy.
>
>
> As expected, the correct symmetry yields the best performance. Our method closely matches this while **automatically discovering the symmetry** (cosine ≈ 1). In contrast, an incorrect symmetry severely degrades both prediction (MSE ↑) and alignment (~0.05), showing that wrong inductive bias is harmful.
>
> The no-symmetry model performs worse than both correct and learned cases, highlighting the benefit of symmetry structure. Overall, our method achieves near-oracle performance without prior knowledge, while avoiding failure modes of incorrect symmetry assumptions.
>
>
>
> > W2:  Additional baseline results
>
> We have already provided comparison with additional baselines such as LieGAN and LieGG in the appendix for angled sine task. For the remaining tasks, please refer the rebuttal to reviewer **uxei-W3**
>
> > W2: Pure symmetry discovery tasks
>
> We interpret “pure symmetry discovery” as methods that focus solely on recovering generators (without learning a predictive model), e.g., LieGAN. Comparisons with LieGAN on the Angled Sine dataset are already provided in the Appendix (Sec. G.3); we summarize the full results here:
>
> | Method | Top-tagging (TT) | Double pendulum (DP) | Angled Sine (AS) |
> |---|---:|---:|---:|
> | LieGAN | **1.000 ± 0.000** | 0.944 ± 0.003 | 0.828 ± 0.138 |
> | $H_\gamma$-Net (e) | **1.000 ± 0.000** | **1.000 ± 0.000** | **1.000 ± 0.000** |
>
> Performance is measured via cosine similarity with the ground-truth generator. Both methods succeed on TT (≈1.0). However, on more challenging settings (DP, AS), LieGAN shows degraded and higher-variance recovery, whereas $H_\gamma$-Net consistently achieves perfect alignment across all tasks.
>
> This indicates that our method is not only competitive with dedicated symmetry discovery approaches, but also more robust, while additionally supporting end-to-end predictive learning.
>
>
> > Q1: On completeness of the classification
>
> Our elliptic(e)/hyperbolic(h)/parabolic(p) split is **not a complete partition of all Lie algebra elements**, but a decomposition into canonical geometric components (rotation, scaling, shear). Generators with mixed 2x2 blocks can indeed arise, but correspond to coupled actions with more complex orbits.(see rebuttal to reviewer **uxei-W1**)
>
> Our framework targets 1D subgroups with a pure (e/h/p) structured component. When a larger group admits a basis whose generators are individually of these types (not necessarily the same), we can recover them **independently** and reconstruct the subgroup.
>
> If a generator is a  mixture of components, our current framework does not apply. Extending to such cases is left for future work.

---

> > ### Author Rebuttal · Reviewer_zEoD · 2026-04-02
> >
> > I thank their authors for their detailed rebuttals and additional experiments.
> >
> > A few additional comments.
> >
> > **W3** I agree that by redefining the domain the functions become continuous, but it just hides potential discontinuity problems. I think it is more accurate to simply state that it seems not to be an issue in practice and adding a discussion of how they can be mitigated by various techniques.
> >
> > I agree with the other reviewers' concern with the interpretability claim. It seems like the authors define it as having an equivariant model; this is not what I would expect as the definition. Similar to other reviewers, I would expect interpretability to mean that it is obvious what symmetry was discovered. I would appreciate a clear definition of what the authors mean by interpretability and justification for that definition.

---

> > > ### Author Response · Authors · 2026-04-03
> > >
> > > > W3
> > >
> > > We thank the reviewer for the insightful comment regarding continuity.
> > >
> > > We agree that restricting to admissible domains does not remove discontinuities. As you suggested, we state that it is not an issue in practice and we will also add a discussion section.
> > >
> > > Motivated by this comment, we further propose another practical mitigation strategy (apart from standard numerical techniques like $\varepsilon$-stabilization or clipping) based on **multiple canonicalization pivots**. Let:
> > >
> > > $$
> > > z_1(x) = \mathrm{invRep}^{(1)}(x), \quad
> > > z_2(x) = \mathrm{invRep}^{(2)}(x),
> > > $$
> > >
> > > where each map is invariant on its admissible domain.
> > >
> > > Their concatenation
> > > $$
> > > z(x) = (z_1(x), z_2(x))
> > > $$
> > >
> > > remains invariant on the intersection of these domains, since for any group element $g$,
> > >
> > > $$
> > > z(g \cdot x) = (z_1(g \cdot x), z_2(g \cdot x)) = (z_1(x), z_2(x)) = z(x).
> > > $$
> > >
> > > To improve robustness near boundary regions, we can use a smooth combination:
> > >
> > > $$
> > > z(x) = \big(m_1(x) z_1(x), m_2(x) z_2(x),  m_1(x),  m_2(x) \big),
> > > $$
> > >
> > > where $m_i(x)$ are smooth confidence weights (e.g., based on pivot magnitudes). This yields a **multi-chart (atlas-like) representation**, where the singular set of one chart need not coincide with that of another, improving stability near boundaries.
> > >
> > > This idea naturally extends to multiple pivots. In general, one can use up to $\lfloor n/2 \rfloor$ blocks, with the practitioner free to choose any number between $1$ and $\lfloor n/2 \rfloor$ for added flexibility. While this increases model complexity, the overhead is negligible relative to the backbone network $\phi$.
> > >
> > > Importantly, this approach does not completely remove singularities globally, but may further reduce practical instability by avoiding reliance on a single pivot.
> > >
> > > We will include this clarification and discussion in the revision.
> > >
> > > > Interpretability Definition
> > >
> > > We thank the reviewer for this valuable feedback. We acknowledge the confusion regarding the term **"interpretability."**  While reviewers associate it with **human-intuitive explanations**, we had **defined** it in this context as **structural transparency** and **analytical identifiability**.
> > >
> > > Unlike methods such as Augerino, LieGG, or BeyondAffine, where symmetry is implicit or empirically verified (e.g., via sampled checks of the learned vector field being orthogonal to the gradient field), in our method the equivariance of the prediction model  is **provably guaranteed** by the internal geometry of the `invRep` layer.  This allows us to precisely pinpoint the architectural source of equivariance.  We agree that terms such as **"analytically identifiable"** or **"structurally transparent"** are more precise and will adopt this terminology in the revision to resolve any ambiguity.
> > >
> > > So in summary, what we mean by interpretability is **mathematical/analytical provability of the equivariance of the prediction model**, not just the interpretability of discovered symmetry. In revised draft, we will use appropriate terminology.
> > >
> > > > Justification of interpretable model (also includes response to reviewer *uxei* rebuttal ack comment):
> > >
> > > We agree that a two-stage pipeline (discover symmetry → design equivariant model) is possible. However, this approach introduces two practical challenges:
> > >
> > > (i) **Model design complexity.** Constructing equivariant architectures for a discovered symmetry is often non-trivial (e.g., EMLP requires solving for appropriate representation spaces), and may be computationally demanding.
> > >
> > > (ii) **Post-hoc discovery limitations.** Methods that first discover symmetry (e.g., LieGG, BeyondAffine) operate over a broad function class and rely on empirical criteria, making symmetry identification itself computationally challenging (see [1]).
> > >
> > > In contrast, our approach integrates symmetry discovery and modeling by restricting the hypothesis class to **analytically invariant/equivariant functions**. This yields:
> > >
> > > - a **mathematically provable equivariant model**,
> > > - no need for a separate architecture design step, and
> > > - improved stability in both symmetry discovery and prediction (as observed empirically).
> > >
> > > Thus, interpretability in our setting refers to **explicit, parameterized symmetry structure directly tied to the model**, rather than post-hoc verification. We will clarify this definition in the revision.
> > >
> > > We acknowledge that this comes with a trade-off in generality, as our method focuses on structured symmetry classes.
> > >
> > > > Extension to separably mixed type and coverage of most generators
> > >
> > > We have provided another extension of our work (based on already existing techniques in our main paper) to separable mixed type. These along with pure types cover most generators (the inseparable ones constitute measure zero set in parameter space). Please refer to our *C1-2* response to reviewer **uxei**'s ack comment.
> > >
> > >  [1] Ensign et al., The Complexity of Explaining Neural Networks Through (group) Invariants, ALT 2017.

---

### Decision · Program_Chairs · 2026-04-30

**Decision:**

Accept (regular)

**Comment:**

Initial reviews were mixed. Reviewers valued the main idea of jointly learning the symmetry and equivariant model, and the experimental results. The major concerns were about potential limitations to certain groups, limited applicability, and questionable interpretability claims.

The rebuttal period was productive and strengthened the paper. The authors clarified the interpretability claims, showed how the method applies to higher dimensional groups, and that less restrictive baselines perform worse on the tasks considered. Two reviewers were convinced and increase their scores to recommend acceptance. I encourage the authors to incorporate the suggested changes in the next revision.

Reviewer ppNx remains unconvinced, mainly concerned with the applicability of the method. Other reviewers had raised similar issues but did not consider them serious enough for rejection. I follow the reviewer majority and recommend acceptance.